# BEHIND THE MAGIC, MERLIM: MULTI-MODAL EVALUATION BENCHMARK FOR LARGE IMAGE-LANGUAGE MODELS

## ABSTRACT

Large Vision and Language Models have enabled significant advances in fully supervised and zero-shot visual tasks. These large architectures serve as the baseline to what is currently known as Instruction Tuning Large Vision and Language models (IT-LVLMs). IT-LVLMs are general-purpose multi-modal assistants whose responses are modulated by natural language instructions and visual data. Despite this versatility, IT-LVLM effectiveness in fundamental computer vision problems remains unclear, primarily due to the absence of a standardized evaluation benchmark. This paper introduces a Multi-modal Evaluation Benchmark named MERLIM, a scalable test-bed to assess the capabilities of IT-LVLMs on fundamental computer vision tasks. MERLIM contains over 300K image-question pairs and has a strong focus on detecting cross-modal "hallucination" events in IT-LVLMs. Our results bring important insights on the performance of state-of-the-art IT-LVLMLs[1AUQ] including limitations at identifying fine-grained visual concepts, object hallucinations across tasks, and biases towards the language query. Our findings also suggest that these models have weak visual grounding, but manage to make adequate guesses from global visual patterns or language biases contained in the LLM component. We name this phenomena of correct answers with no visual grounding as hidden hallucinations[4cSE]

## 1 INTRODUCTION

Large-scale transformer networks have significantly advanced the field of Natural Language Processing (NLP) (Brown et al., 2020; Devlin et al., 2018; Liu et al., 2019; Raffel et al., 2020). Large Language Models (LLMs) have demonstrated remarkable zero-shot performance in a wide range of NLP tasks (Lin et al., 2022; Min et al., 2023), and their success has resulted in a fundamental shift for NLP research, moving from task-specific pre-training to task-agnostic representation learning. Similarly, in the image domain, vision transformers trained on large-scale image collections have achieved state-of-the-art performance on multiple fully supervised tasks (Dosovitskiy et al., 2020; Fan et al., 2021; Liang et al., 2021). The integration of a language stream has also demonstrated empirical benefits in fundamental computer vision tasks such as recognition (Alayrac et al., 2020; Jia et al., 2021; Radford et al., 2021) and detection (Chen et al., 2021; Du et al., 2022). Ultimately, these advances in computer vision and NLP fields have converged in the development of Large Vision and Language models (LVLMs) (Alayrac et al., 2022; Kim et al., 2021; Li et al., 2022a; Nichol et al., 2021; Radford et al., 2021; Yu et al., 2022), which have achieved state-of-the-art performance on several downstream visual tasks while enabling significant improvements in the zero-shot setups.

Following the success of these large-scale multi-modal transformers, a new trend has emerged which involves integrating frozen LLMs and vision transformers by means of a smaller neural network (Bai et al., 2023; Dai et al., 2023; Dong et al., 2024; Huang et al., 2023b; Li et al., 2023a; Liu et al., 2023; Peng et al., 2024; Xue et al., 2024; Tsimpoukelli et al., 2021; Zhai et al., 2022; Zhu et al., 2023). This straightforward modification only requires the training of a fusion module to interface between the frozen modality encoders. The resulting model requires significantly less computational effort to train, and enables querying image data with natural language, thus providing a contextualized natural language response through the frozen language decoder (Tsimpoukelli et al., 2021).

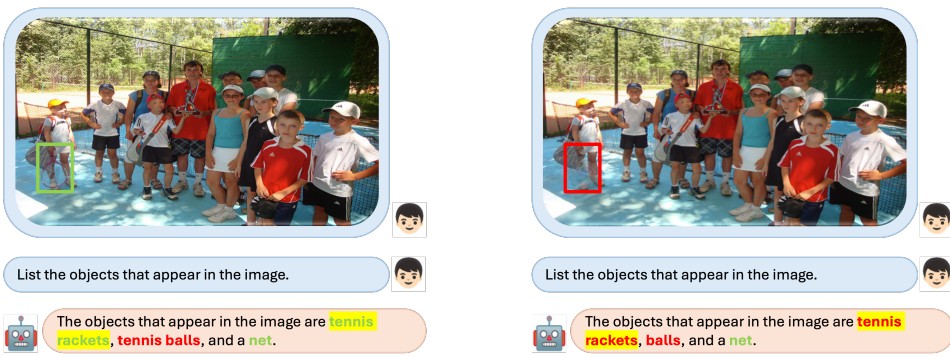

Figure 1: **IT-LVLM "hidden" Hallucinations**. On the left sub-figure we show the prediction of InstructBLIP  Liu et al. (2023), despite the small size of the tennis racquet and the relatively cluttered scene, we get an (apparently) correct response including the racquet object (we highlight in green font true positives and in red font false positives). After a subtle image edit (right sub-figure), the response remains the same although there is no visual grounding for the tennis racquet. The box around the racquet was drawn for easier visualization, but it is not part of the original visual input.

Such models are known as Instruction Tuning Large Vision and Language models (IT-LVLM), as their zero-shot capabilities are mediated by a natural language input that instructs the model about the expected output (Huang et al., 2023b). Although IT-LVLMs are intended to be general-purpose instructional assistants, there is still no standard test-bed to assess their zero-shot effectiveness in fundamental computer vision tasks. The presence of an LLM introduces additional challenges, among them the inherited tendency to Hallucinate (Huang et al., 2023a; Zhang et al., 2023), resulting in the emergence of language outputs that can not be grounded to the visual input.

In this paper we take an in depth look at the capabilities of IT-LVLMs for fundamental computer vision tasks, while assessing the impact of hallucination events in the model's performance. Our proposal has two key elements: First, instead of presenting the model with a fixed set of possible answers, we allow the IT-LVLM to reply in natural language and then parse the open-set IT-LVLM prediction. Second, to better analyze the hallucination issue, we propose an inpainting procedure that subtly manipulates both the image and its corresponding ground-truth. This technique enables us to identify two types of hallucinations: (1) Regular Hallucinations, which are outputs that do not correspond to the image, and (2) Hidden Hallucinations, which appear to be true positives but lack visual grounding. We illustrate these two phenomena in Figure 1, where an ostensibly correct prediction, such as "Tennis Racquet," lacks visual grounding in the edited image, thereby categorizing it as a "hidden hallucination".

We compile our image edits, corresponding ground-truths, and parsing strategy into MERLIM (**M**ulti-modal **E**valuation benchma**R**k for **L**arge **I**mage-language **M**odels), a novel test-bed aimed at empirically evaluating IT-LVLMs on core computer vision tasks including object recognition, instance counting and identifying object-to-object relationships. Our benchmark serves as a straightforward and extensible evaluation strategy of core computer vision tasks on well-established image datasets like MS-COCO (Lin et al., 2014), LVIS (Gupta et al., 2019), and Visual Genome.

Our paper brings the following contributions: **(i)** A standardized test-bed to assess the zero-shot effectiveness of IT-LVLMs on fundamental computer vision tasks, including Object Recognition, Object Counting, and Inter-object Relationship Understanding. **(ii)** Through extensive empirical evaluation, we identify and quantify "hidden" hallucination errors which are out of the scope of other benchmarks, we observe this hallucination unfairly increase the accuracy of IT-LVMLMs. **(iii)** We conducted a thorough analysis of language biases through extensive experiments using a diverse set of carefully selected question prompts. Our observations reveal that IT-LVLMs exhibit a strong bias toward the language tokens of both the question and answer. MERLIM's code is available here

## 2 RELATED WORK

The widespread adoption of LLMs and their zero-shot capabilities (Brown et al., 2020; Chiang et al., 2023; Devlin et al., 2018; Liu et al., 2019; Raffel et al., 2020) has led to an increased interest in the

development of LVLMs (Jia et al., 2021; Pham et al., 2023; Radford et al., 2021; Yu et al., 2022; Zhai et al., 2022). LVLMs typically build upon metric learning, estimating a similarity measure across modalities, making these architectures effective in inherently multi-modal tasks such as VQA and text-image retrieval.

**Visual Instruction Tuning**    The effectiveness of LVLMs, coupled with the unprecedented availability of large-scale data in the visual domain, has led to the development of IT-LVLMs. Unlike LVLMs, IT-LVLMs allow to prompt a joint language and visual representation and then generate results with a frozen language decoder. This procedure preserves the multi-modal capabilities of the LVLM, and enables the use of natural language queries as an instruction set that modulates the expected output (Alayrac et al., 2022; Tsimpoukelli et al., 2021).

A pioneer work exploring the few-shot learning abilities of frozen language models and vision was presented by Tsimpoukelli et al. (2021) where the image input is encoded into a feature set and inserted as a prefix in the text stream, thereby conditioning the responses of the language network. Flamingo (Alayrac et al., 2022) introduced an auto-regressive text model conditioned on visual input, by using gated cross-attention. The tokens from the perceiver architecture (Jaegle et al., 2021) are fed into a frozen language model. This Gated cross-attention was extended by BLIP-2 (Li et al., 2023a), linking the T5 language model and its visual stream (Raffel et al., 2020) with the Q-Former module, thus improving the modality alignment at training time. Recent work has continued to explore the visual-to-language alignments on IT-LVLMS, either by improving the training and annotation data (Liu et al., 2023; Zhu et al., 2023), or by introducing large-scale data on the training of the IT-LVLM (Bai et al., 2023; Dai et al., 2023).

**VQA Benchmarks & Datasets**    Many existing benchmarks and datasets for IT-LVLMs are designed for the standard VQA tasks, including a pre-defined set of questions that can be answered by looking at image or video data (Antol et al., 2015; Chen et al., 2015; Fu et al., 2023; Goyal et al., 2017; Gurari et al., 2018; Hudson & Manning, 2019; Zhao et al., 2022). Although these benchmarks provide a direct evaluation of the multi-modal capabilities of LVLMs, they do not assess if the reply originated from the global visual context of the image or from the local visual patterns that effectively ground the prompted question. In contrast, MERLIM focuses on querying and attributing the response of the IT-LVLM to a specific set of object instances in the image by means of small image alterations and direct verification of the same query on the modified visual data. Moreover, we explore the connection between these replies (original image and edited image) and the potential hallucinations from the LLMs component, as some responses can only be attributed to the LLM hallucinating objects.

Perhaps the most similar benchmarks to ours are POPE (Li et al., 2023b) and NExT-GQA (Xiao et al., 2023), as both these benchmarks aim at finding direct visual groundings related to the language query. While POPE proposes to prompt for the presence/absence of a single instance to evaluate hallucinations (in the form of a false positive), we directly modify the visual ground-truth and can verify if the predictions have an effective visual grounding. NExT-GQA approximates the grounding of a prediction by the accurate temporal localization of the event. Despite the straightforward proposal, contemporary research (Otani et al., 2020) has shown that temporal action localization commonly suffers from biased predictions, in particular, this bias emerges directly from the language stream.

## 3   MERLIM

MERLIM's main goal is to assess the effectiveness of Instruction-Tuning Large Visual Language Models (IT-LVLM) in fundamental computer vision tasks, while simultaneously evaluating if the responses have an effective visual grounding. In practice, we design an evaluation scheme that queries the IT-LVLM with a set of predefined language questions. Then, we parse the text responses, transforming the natural language output into a suitable prediction format for each task. This scheme enables us to measure the performance of the IT-LVLM using standard image evaluation metrics, such as accuracy, precision, recall, and F1 metrics. Across all tasks, MERLIM contains a total of 300,664[1][AUQ] unique image-question pairs. Table 1 outlines the sizes of the query set and the image data for all the selected tasks.

**Notation & Definitions.**    A pre-trained IT-LVLM ($h$) takes as input an image ($i$) and a language query ($l$), producing a language response ($r$). Formally: $r = h(f(i, \theta_f), g(l, \theta_g), \theta_h)$, where $f$ is

Table 1: **MERLIM Benchmark.** We incorporate three core visual tasks to benchmark IT-LVLMs: Object Recognition, Object Counting, and Relationship Understanding. We leverage images from the validation set of MS-COCO (Lin et al., 2014) and create edited versions of these images ("Edited" column). Moreover, we extend the MS-COCO ground-truth classes by incorporating the class labels present in LVIS (Gupta et al., 2019) and include the relationships proposed in the overlapping set of Visual Genome (Krishna et al., 2017).

| Task | Dataset | | | | | | |
|------|---------|--|--|--|--|--|--|
| | Response Type | Prompts Per Image | Images | | Labels | | |
| | | | Original | Edited | MS-COCO | LVIS | V. Gnome |
| Object Recognition | Set/Enumeration | 5 | 4183 | 31373 | ✓ | ✓ | ✗ |
| Object Counting | Numeric | 2 | 4183 | 31373 | ✓ | ✗ | ✗ |
| Relationship Underst. | True / False | 4 | 1240 | 5630 | ✓ | ✗ | ✓ |

an image encoder with parameter set $\theta_f$, $g$ is a language encoder with parameter set $\theta_g$, and $\theta_h$ are IT-LVLM specific parameters that integrate the two modalities. In MERLIM, we define a fixed language query set $(L)$, where $l \in L$, and transform the open set output $r$ into a set $r' \in K$ where $K$ can be directly matched to the closed-set ground truth of fundamental computer vision tasks.

**Datasets and Tasks.** We focus our study on three core computer vision tasks: **(i)** Object Recognition, **(ii)** Object Relationship Understanding, and **(iii)** Object Counting. The primary source of data for our Benchmark is the validation set of MS-COCO, where we can directly evaluate the Object recognition and instance counting. For the Reasoning Task, we resort to the labels provided on the validation set of Visual Genome (Krishna et al., 2017).

Considering the open nature of the vocabulary on IT-LVLMs responses, some correctly grounded visual objects could be labeled as false predictions if their class name is not included in the 80 Categories of MS-COCO ground truth labels. To alleviate this undesired penalty, we complement the MS-COCO ground-truths with the labels provided by the LVIS dataset (Gupta et al., 2019). This extended ground-truth allows us to compare against a richer vocabulary.

**IT-LVLM Hallucinations.** In MERLIM, we assess the effective visual grounding of the IT-LVLM responses through two separate tests. First, we directly compare the responses against the dataset ground truth to identify *Regular Hallucinations*, which occur when the predicted noun is not present in the image ground truth. Second, we edit images such that one of the object instances disappears and then compare the IT-LVLM responses when presented with the original $(i)$ and edited $(\hat{i})$ versions. This allows us to identify *Hidden Hallucinations*, which correspond to correct predictions that lack any visual grounding. These errors are beyond the scope of any other multi-modal hallucination benchmarks like Li et al. (2023b); Xiao et al. (2023). Figure 1 illustrates this process: the original image (left) includes a tennis racquet (near the leg of the child located on the far left), while in the edited version (right), the tennis racket was inpainted. Despite the absence of the racquet, InstructBLIP with Vicuna7B continues to infer its presence. To minimize the alterations to the original image, we remove only one object instance at a time. We use the inpainting method proposed by Li et al. (2022b) to fill in the ground-truth segmentation mask according to the image context. The edited validation set expands to a total of 31373 inpainted images obtained from the 4183 original images in MS-COCO val. We empirically verify that our edits do not significantly alter the global or local appearance of the image $(\hat{i})$, we include a detailed analysis of this evaluation in the dTrJ **Supplementary Material**.

### 3.1 OBJECT RECOGNITION

We evaluate the object recognition capabilities of IT-LVLMs on the MS-COCO dataset by adapting its ground-truth for the multi-label classification task. If at least one instance of class $c$ is grounded in image $i$, that indicates the image contains the class $c$, and is therefore expected in $r$. In this task, we design a query set for the IT-LVLM requesting a list of all the visible objects in the image.

**Prompt Construction.** To further analyze the bias towards a particular language prompt, we formulate 5 similar prompts that instruct the IT-LVLM to list the objects present in the images. Our set $P$ for the object recognition task is detailed in the following listing:

1. "Itemize the things seen in the image".    2. "Identify the objects within the image".
3. "Detail the items shown in the picture".    4. "List the objects that appear in the image".
5. "Enumerate the items visible in the picture".

**Evaluation.** We identify all the nouns in $r$ and compare them against the ground truth labels of MS-COCO. To do this, we split the sentence by words and determine if it is a noun or not using the spaCy library (Honnibal & Montani, 2017). We discard words that refer to places or areas like room, street, square, etc. We consider compound nouns as one, for instance, "the back of the car" and "ice cream". We designate a prediction as a True Positive if the predicted word matches or is a synonym of a ground truth element. Likewise, a prediction that does not match any ground truth label or any of its synonyms is labeled as a False Positive. Finally, any ground truth class that can not be found in the prediction (including their synonyms) is regarded as a False Negative. These definitions enable us to calculate precision, recall, and F1 metrics for all IT-LVLMs.

To better manage the extensive vocabulary output of IT-LVLMs, we also utilize the WordNet hierarchy (Miller, 1995), and the API provided by ChatGPT (OpenAI, 2023) to identify potential synonyms for elements in $r$. We pre-compute a corpus of noun-to-noun relations totaling 334608 tuples and stored them, therefore future changes in ChatGPT model/API will not affect our benchmark predictions. On occasions, we discover unsuitable outputs, *e.g.*, "There are many objects" or "It is a beautiful scene", the recall and precision for such responses are defined as 0. For completeness, we also report the amount of such instances in the ᵈᵀʳᴶ **Supplementary Material**.

### 3.2 Inter-object Relationship Understanding

After evaluating the object recognition capabilities, we proceed with a finer-grained task, and evaluate the capability of these models to infer simple inter-object relationships. To this end, we use the subset of COCO images that overlaps with the Visual Gnome labels. On this subset, we query the model about object relationships with visual grounding in the original ($i$) and edited images ($\hat{i}$).

**Prompt Construction.** Since the IT-LVLM outputs are much harder to constrain and parse in this task, we only formulate questions with a yes or no response. We also verify if the described inter-object relationship has a visual grounding or not. For instance, for the images in Figure 1, we could query the model about the relationship between the tennis racquet and the child: *"Does the child have a tennis racquet?"*. Since Visual Genome does not provide, questions with this exact structure, we use the ChatGPT API (OpenAI, 2023) to build a natural language question with a Yes/No answer:

**Authors:** *"Correct the mistakes in the following question: 'skateboard has wheels?' Please, just write the correct question in present tense and consider it is a yes or no question"*.
**ChatGPT API:** "Does the **skateboard have wheels**?"

**Relationship Ambiguity.** It is challenging to direct the IT-LVLM attention to a specific instance when there are multiple objects of the same class. This makes the visual grounding assessment ambiguous, as the target relationship might randomly exist among two other entities with the same semantic classes. We mitigate this ambiguity by substituting nouns that have two or more instances in the image with another class noun. We follow two strategies to select the new object categories: (1) We randomly sample another object category from the ground truth classes of MS-COCO which are not included in the image ground truth. We name this set the *Random Set*. (2) We ask the ChatGPT API (OpenAI, 2023) to select an object category to generate a meaningful relationship (in terms of language). This set of relationships is named the *Curated Set*. The expected answer to the queries in any of these sets are always negative, as the relationship no longer has a valid visual grounding.

The main difference between the random and curated sets is that, the first one can generate relationships that do not make sense and can be trivially evaluated using an LLM, for example, *"Does a clock have wheels?"*. Meanwhile, the curated set has more plausible relationships, which are better suited for multi-modal data, for example: *"Does the water bottle have a sticker?"*. We provide the results

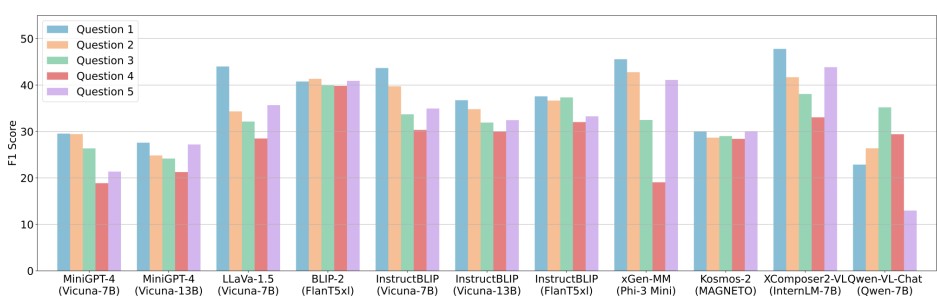

Figure 2: **Results for the description task.** We report the F1 Score on the original images with the set of five prompts. We highlight that BLIP-2 and Kosmos-2 are the only IT-LVLMs that obtain a similar performance across the five prompts. The best performance for any prompt is attained by InternLM-XComposer2-VL and xGen-MM, critically these models exhibit 14.73% and 26.50% variability between their best score and the worst, respectively.

of both sets to better analyze the influence of language biases, and possible uni-modal (language only) shortcuts taken by the IT-LVLM. Finally, we also evaluate the response for a negative version of the query. Again, we use the ChatGPT API to generate negative questions based on the original queries. For instance, *"Does a bicycle have wheels?"* is the negative form of *"Does a bicycle not have wheels?"*.

**Evaluation.**  Since the model's answers are restricted to Yes / No questions, we compute the accuracy (Acc) of the IT-LVLMs responses. Furthermore, we compare the accuracy between the original and edited image sets to assert how much the model's answers rely on robust visual grounding.

## 3.3  OBJECT COUNTING

Finally, we also evaluate the counting capabilities of IT-LVLMs. Although this task is common in VQA benchmarks, even state-of-the-art methods struggle to provide accurate results. In particular, the counting task requires the models to have strong object recognition and instance detection capabilities that enable them to count in a discrete fashion (Trott et al., 2018; Zhang et al., 2018).

**Prompt Construction.**  We instruct the model to count the objects in the original and the edited images and require its response to be strictly a number. Unlike other tasks, we share a single prompt for the entire set [1AUQ] *"How many **object_category** are there? Just answer the number."*. As a second step, we check the consistency of the answers. We formulate yes or no questions based on previous answers over the very same image. For instance, if the model responds *"There are **five** books"*, on the second step we prompt: *"Are there **five** books?"* using the very same image.

Although the task is very challenging, we focus on the consistency exhibited by the IT-LVLM on the follow-up questions, and the responses obtained from the edited images (which should have exactly 1 less object), even if the actual number of objects does not match the ground-truth, it would be expected for the secondary query to be consistent with the initial one. This query scheme helps us understand if the IT-LVLM leverages the same visual patterns despite being queried in syntactically different ways.

**Evaluation Scheme.**  As we instruct the model to deliver a numeric answer, we compute the mean and standard deviation of the absolute error from the original and edited sets. Additionally, we calculate the accuracy metric for secondary questions designed to return Yes/No responses.

## 4  MERLIM RESULTS

We proceed with the experimental assessment of several representative state-of-the-art Instruction Tuning Large Visual Language Models, following the protocol outlined in MERLIM. Specifically, we evaluate BLIP-2 (with FlanT5xl LLM) (Li et al., 2023a), InstructBLIP (with Vicuna-7B v1.1, Vicuna-13B v1.1 and FlanT5xl LLMs) (Dai et al., 2023), MiniGPT-4 (with Vicuna-7B v0 and Vicuna-13B v0 LLMs) (Zhu et al., 2023), LLaVA-1.5 (with Vicuna-7B v1.5 LLM) (Liu et al., 2023), xGen-MM (with

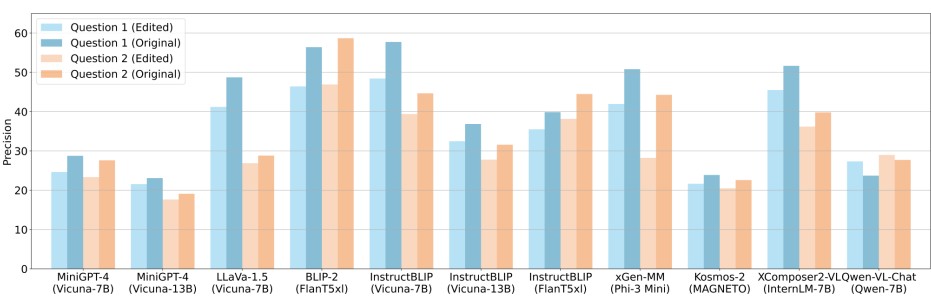

Figure 3: **Results on the original and edited sets.** We compare the precision of IT-LVLMs on the edited and original image sets across two prompts. To analyze the hidden hallucination problem, we focus on the subset where the image inpainting removes an entire category. Notably, all methods lose performance on the edited image set.

Phi-3 Mini 3.8B LLM) (Xue et al., 2024), Kosmos-2 (with MAGNETO LLM) (Peng et al., 2024), InternLM-XComposer2-VL (with InternLM-7B LLM) (Dong et al., 2024), and Qwen-VL-Chat (with Qwen-7B LLM) (Bai et al., 2023). We outline and analyze the results across the three evaluation tasks in MERLIM.

### 4.1 OBJECT RECOGNITION

IT-LVLMs aim to deliver coherent, detailed, and visually grounded language responses based on the provided images and questions. To assess their effectiveness, we first evaluate the performance of all IT-LVLMs reporting the F1 Score in MERLIM, for each of the five proposed questions. Then, we take a closer look at the precision metric, as it is closely linked to hallucinations events, a lower precision indicates a higher rate of false positives due to object hallucinations. We also perform a detailed analysis for a subset of MERLIM, to quantify Hidden Hallucination events. In this subset, the inpainting strategy has fully removed the visual grounding of an entire category (*i.e.* there was a single instance of the category). As outlined in section 3, we compare the response on the original image and the edited data to discover hidden hallucinations.

**Instruction Bias.**  Figure 2 outlines the performance (F1 score) of the IT-LVLMs in MERLIM for each prompt. Interestingly, while ChatGPT (OpenAI, 2023) and Vicuna-33B (Chiang et al., 2023) consider the prompts semantically identical, only BLIP-2 (Li et al., 2023a), and Kosmos-2 (Peng et al., 2024) exhibit consistent performance across all five instructions. These models boast low standard deviations of 0.59 and 0.67, respectively. Notably, BLIP-2 achieves the best average performance with 40.55%, even outperforming its successor, InstructBLIP. This superior consistency of BLIP-2 over InstructBLIP might stem from its lacking an instruction-aware Q-former module. This module's presence in InstructBLIP could make it more susceptible to slight variations in the phrasing of instructions.

The best performance for a single question is attained by InternLM-XComposer2-VL on question 1 with an F1 Score of 47.79%. In contrast, MiniGPT-4 (Vicuna-13B) exhibits the lowest average performance with 25.02%, which might be attributed to the version (v0) of Vicuna and the quality of its multimodal instructional training data.

**Object Hallucination.**  MERLIM can break down Hallucination events into regular and hidden Hallucinations. To this end, we compare the precision of IT-LVLMs on both the original and edited image sets. Figure 3 presents results for two of the five designed prompts[1]. Regarding regular hallucinations, 5 of the 11 models we tested InstructBLIP$^{\text{dTrJ}}$ (Dai et al., 2023) InternLM-XComposer2-VL$^{\text{dTrJ}}$ (Dong et al., 2024), BLIP-2$^{\text{dTrJ}}$ (Li et al., 2023a), LLaVA-1.5$^{\text{dTrJ}}$ (Liu et al., 2023) and xGen-MM$^{\text{dTrJ}}$ (Xue et al., 2024) achieved a precision near or above 50% on the original image set in at least one of the selected questions. Surprisingly, these models exhibit the largest performance gap on the edited image set, with an average drop of 8.05%. This empirical evidence indicates that the best performing methods are also the ones which suffer the most from hidden hallucinations.

---

[1]We chose Questions 1 and 2 as they are, on average, the best-performing prompts by F1 Score

Table 2: **Results for Inter-object Relationship Understanding.** We report the Accuracy (Acc) of the IT-LVLMs in the two sets: one with randomly sampled relationships (Random Set) and another where the relationships are curated by an LLM (Curated Set). We compute the Accuracy with the affirmative (**Acc**) and negative (**Acc$^{neg}$**) versions of the relationships to validate the models' answers. We also present the Acc in both sets of COCO images, the edited (**Acc$_{ed}$**) and Original (**Acc$_{org}$**), and the absolute accuracy difference $\mathbf{\Delta Acc = Abs(Acc_{org} - Acc_{ed})}$. The Curated set constitutes a more challenging set for all methods because. No IT-LVLM is superior across all evaluated scenarios.

| Model | LLM | Random Set | | | | | Curated Set | | | | |
|---|---|---|---|---|---|---|---|---|---|---|---|
| | | Acc$_{org}$ ↑ | Acc$_{ed}$ ↑ | $\Delta$Acc ↓ | Acc$_{org}^{neg}$ ↑ | Acc$_{ed}^{neg}$ ↑ | Acc$_{org}$ ↑ | Acc$_{ed}$ ↑ | $\Delta$Acc ↓ | Acc$_{org}^{neg}$ ↑ | Acc$_{ed}^{neg}$ ↑ |
| MiniGPT-4 | Vicuna-7B v0 | 41.13% | 37.35% | 3.78% | 23.77% | 20.67% | 37.38% | 36.77% | **0.61%** | 29.29% | 22.36% |
| MiniGPT-4 | Vicuna-13B v0 | 45.35% | 44.81% | **0.54%** | 34.30% | 31.87% | 41.01% | 41.81% | 0.80% | 41.06% | 34.28% |
| LLaVA-1.5 | Vicuna-7B v1.5 | 55.59% | 40.37% | 15.22% | 68.85% | 69.00% | 62.06% | 24.87% | 37.19% | 83.26% | 80.18% |
| BLIP-2 | FlanT5xl | 83.62% | 78.61% | 5.00% | 48.80% | 49.38% | 67.84% | 62.04% | 5.80% | 49.77% | 45.88% |
| InstructBLIP | Vicuna-7B v1.1 | 91.17% | 86.27% | 4.90% | 19.60% | 9.91% | 76.75% | 67.87% | 8.88% | 45.31% | 16.34% |
| InstructBLIP | Vicuna-13B v1.1 | 90.32% | 86.98% | 3.34% | 16.24% | 5.58% | 75.99% | 71.49% | 4.49% | 43.52% | 12.17% |
| InstructBLIP | FlanT5xl | 80.69% | 75.10% | 5.59% | 77.16% | 79.49% | 70.15% | 57.19% | 12.95% | 63.90% | 70.98% |
| xGen-MM | Phi-3 Mini 3.8B | 61.31% | 46.45% | 14.86% | 49.61% | 49.06% | 66.36% | 32.89% | 33.46% | 64.41% | 54.48% |
| Kosmos-2 | MAGNETO | 19.34% | 5.33% | 14.02% | **90.47%** | **92.73%** | 45.98% | 4.42% | 41.56% | **85.05%** | **93.98%** |
| XComposer2-VL | InternLM-7B | 71.47% | 60.32% | 11.15% | 43.91% | 51.60% | 72.96% | 45.67% | 27.30% | 48.69% | 63.37% |
| Qwen-VL-Chat | Qwen-7B | **92.11%** | **88.17%** | 3.94% | 15.81% | 3.34% | **81.41%** | **76.79%** | 4.63% | 42.29% | 4.65% |

Clearly some replies lack direct visual grounding (as the corresponding visual instance was removed from the image in the inpainting process), however these objects persist in the language prediction. This result suggests that the best performing IT-LVLMs are obtaining an advantage by prioritizing the global visual context or previously predicted language tokens over the effective visual grounding. We provide further evidence to support this observation in Table 5 $^{dTrJ}$ **Supplementary Material**. In this table, we show that the gradient contributions of the previously predicted language tokens are, on average, larger than those of the image. This suggest that top-performing models are improving their performance by hallucinating objects which are largely determined by the initial answer tokens.

Additional results for the Precision and Recall metrics are reported in the $^{dTrJ}$ **Supplementary Material**.

## 4.2 INTER-OBJECT RELATIONSHIP UNDERSTANDING

Following Section 3.2, we continue by evaluating the capabilities of IT-LVLMs to identify fine-grained inter-object relationship using the subset of MS-COCO that overlaps with Visual Gnome as described in Table 1. We query the IT-LVLMs about the relationships in the original and edited image sets. Table 2 summarizes the accuracy in both sets ($\mathbf{Acc_{orig}}$) and ($\mathbf{Acc_{ed}}$) respectively. Following the described protocols to mitigate the relation ambiguity in Section 3.2, we also report the accuracy of the models in the two sets of relationships. The Random Set (negative relationships are built from any two random objects) and the Curated Set (we verify the relationship is plausible according to an LLM). Finally, we also query the models with the negative version of the two sets of relationships ($\mathbf{Acc^{neg}}$).

**Performance in the edited images set.** Table 2 shows that all tested IT-LVLMs exhibit an average performance drop of 7.49% when transitioning from the original to the edited set. This performance gap, is nearly the same gap in the object recognition task, suggesting that IT-LVLMs may also be hallucinating certain visual relationships due to the global visual context and the inherent biases within the language models. MiniGPT-4 (Zhu et al., 2023) (with Vicuna-7B or 13B) attains the lowest accuracy overall, although it compensates with the smallest variability between the original and edited sets ($\mathbf{\Delta Acc}$). Despite MiniGPT-4's LLM and Visual stream being comparable in size to other models, the reduced performance could be attributed to a different Vicuna version (V0 for MiniGPT-4) and a relatively small fusion module ($\theta_h$), consisting of a single linear layer. It is noteworthy that no IT-LVLM consistently outperforms the others in all the evaluated scenarios. However, Qwen-VL-Chat (Bai et al., 2023) exhibits the best performance for affirmative relationships in both original and edited images, regardless of the set (random or curated).

**Querying with more realistic relationships.** The curated set requires the IT-LVLMs to build a better understanding of visual information to reply about the visual relationships. In short, relationships can not be trivially resolved from the language query (see Section 3.2). As summarized in Table 2, all the evaluated IT-LVLMs significantly reduce their performance in the edited set by 11.63% when

compared to the curated set. Furthermore, the average performance gap between the original and edited sets ($\mathbf{\Delta Acc}$) has risen to 16.15% more than double the gap observed in the random set.

These results support our hypothesis that some visual relationships are entirely resolved by the language stream, thus artificially increasing the performance of the IT-LVLM. More importantly, visual object relations follow a similar hallucination pattern as the object recognition task, where most of IT-LVLMs favor shortcuts over the language modality, rather than building grounded multimodal representation where inter-object relationship can be modeled and queried.

**LLMs vs IT-LVLMS.** To get a better perspective on the relevance of language shortcuts on the IT-LVLM input, we conduct a comparative analysis of the performance between IT-LVLMs and LLMs (no visual input) in the Inter-object Relationship Understanding task. We observe that LLMs lag behind IT-LVLMs, however, LLMs can outperform a random baseline despite having no visual data which further suggest language only biases constitute a significant party of the performance of the IT-LVLM. We include the detailed analysis in the $^{\mathsf{dTrJ}}$ **Supplementary Material**.

### 4.3 OBJECT COUNTING

In Figure 4, we report the mean absolute error (MAE) and its standard deviation for all tested IT-LVLMs on both the edited (blue) and original (orange) in MERLIM. All IT-LVLMs exhibit a significant increase in error on the edited set. This indicates that the models' intrinsic error in the counting task is not correlated with the visual grounding, but rather a lack of visual grounding in their responses, leading to hallucinations.

**Answer Consistency.** To further investigate language biases, we also compute the accuracy metric for the yes/no questions, which validate the answers given by the IT-LVLM in the initial prediction (*i.e.* Are there N objects?). We highlight that all methods still perform better in the original image set, especially the largest IT-LVLMs, like InstructBLIP over Vicuna-13B. However, the drop in performance remains significant in the edited set. LLaVA (Liu et al., 2023) exhibits the highest consistency, with over 80% of the replies confirming the initial assessment (regardless if the count was accurate or not) for both the edited and original sets. Similarly, the LLaVA IT-LVLM also has the smallest average error and the smallest standard deviation on the initial counting, clearly outperforming every other IT-LVLM in this particular task.

## 5 DISCUSSION & CONCLUSIONS

**Hidden Hallucinations.** Unlike previous benchmarks, MERLIM stands out for its ability defining and studying a novel type of hallucination: hidden hallucinations. These are more challenging to detect because they are seemingly correct predictions but in fact lack a meaningful visual grounding. Our findings suggest that top-performing models might achieve higher scores (in part) due to these

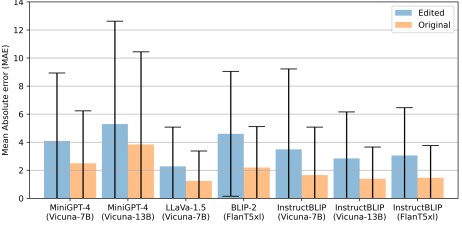 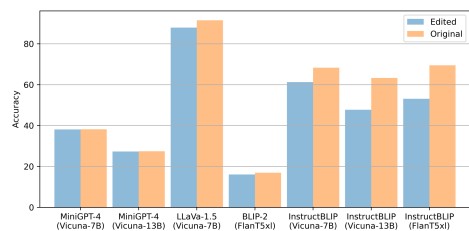

Figure 4: **Results on Object Counting task.** Subfigure a) reports the mean absolute error of IT-LVLMs in the original and edited sets. LlaVa v1.5 Liu et al. (2023) demonstrates better performance across both sets. On average, BLIP2 and MiniGPT make the worst estimations. Every model reduces its performance in the edited set, indicating a weak correlation between visual grounding and model response. Subfigure b plots the accuracy for the yes/no questions designed to validate answer consistency for the main counting task. LlaVa exhibits remarkable consistency across both sets, significantly outperforming other methods. Other IT-LVLMs perform close to random chance, suggesting a bias towards the question format in their answers.

hidden hallucinations. Moreover, our analysis also identifies potential causes for this phenomenon: language biases and a lack of robust and detailed visual grounding.

**Prompt Selection.** Despite current advances in LLMs, slight syntactic changes of input prompt heavily condition the language output of the IT-LVLM. In our benchmark most models vary drastically their outputs and associated performance. Although our queries are semantically equal, all the questions had a different average length, and the average length for the same question would also differ across models. We conclude IT-LVLMs responses remain heavily biased towards the selected query set. To foster future research, we will release the queries and results in MERLIM, to encourage the community to further explore the strong discrepancies observed in this paper.

**Limitations.** MERLIM comprises a total of 300664 unique image-question pairs across all tasks, which may limit its usability for evaluating and prototyping new models. Therefore, we recommend using the subset of MERLIM designed for the Object Recognition Task, where image inpainting removes an entire category. This subset, used in Figure 3, contains 5608 edited and 3037 original images.

**Conclusion.** This paper introduces MERLIM, a novel multi-modal Benchmark for evaluating IT-LVLMs in fundamental computer vision tasks. Despite the versatility of current IT-LVLMs, we observe an overall poor performance in these fundamental tasks, with multiple failure scenarios associated with weak or non-existent visual grounding of the language predictions. We think MERLIM can be the initial effort to better diagnose Hallucination events, and to create stronger IT-LVLMs where ensuring a solid visual grounding could improve the overall quality of the predictions.

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

# A  SUPPLEMENTARY MATERIAL

To complement the experimental assessments in Section 4, we provide a summary of comparisons across a subset of tasks and evaluation metrics in Figure 9. This is followed by an in-depth analysis of the effects of our inpainting rationale and a direct comparison between IT-LVLMs and their LLM-only counterparts in MERLIM. Additionally, we offer insights into language biases by examining the average number of nouns generated for each of the five proposed prompts in Section 3.1. Lastly, we delve into the object hallucination issues by presenting the full Precision and Recall metrics for the models across the original image set, using the same five prompts from Section 3.1. Additionally, we analyze the gradient percentage corresponding to each input (Image, Question, and Answer) to understand their contributions to the generated answers.

## A.1  ANALYSIS OF INPAINTED IMAGES

To analyze the visual changes in the images after the inpainting process, we first assess that the selected inpainting strategy induces a minimal change in the global image features and that the inpainted masks effectively hide the object instance from object detectors. We use YOLOv7 Wang et al. (2023) to verify the absence of the object in the edited image and ResNet50 He et al. (2016) to evaluate the global feature similarity with the original image. For each image in MERLIM, we enforce two essential criteria: (1) YOLOv7 must not generate a detection box with an Intersection over Union (IoU) greater than 0.7 relative to the ground-truth bounding box of the removed instance, and (2) the cosine similarity between the ResNet50 features of the edited image and the original image must exceed 0.7.

For a more in-depth assessment of the visual changes in the inpainted images, we use another box detector, Mask R-CNN He et al. (2017), to verify if each predicted box in the inpainted image has a corresponding box with the same class and a similar spatial location (i.e., high Intersection over Union) in the original image.

We perform this analysis for bounding boxes with confidence scores above 0.7 and 0.5, summarizing the results in Table 3.

Table 3: **Analysis of Inpainted Images.** We analyze the Intersection over Union (IoU) of each box predicted by Mask R-CNN in the edited image set of MERLIM. Table a) presents the analysis for bounding box predictions with a confidence score greater than 0.7, while Table b) shows the analysis for box predictions with a confidence score greater than 0.5. We find that up to 3290 boxes, representing 0.7% of the total, do not exhibit a high overlap with the original boxes.

| IoU | Percentage | # of boxes |
|---|---|---|
| above 0.9 | 85.37% | 259454 |
| 0.8-0.9 | 9.3% | 28250 |
| 0.7-0.8 | 3.1% | 9428 |
| 0.6-0.7 | 1.4% | 4260 |
| 0.5-0.6 | 0.44% | 1324 |
| bellow 0.5 | 0.38% | 1165 |

(a)

| IoU | Percentage | # of boxes |
|---|---|---|
| above 0.9 | 80.68% | 360595 |
| 0.8-0.9 | 10.97% | 49054 |
| 0.7-0.8 | 4.49% | 20097 |
| 0.6-0.7 | 2.23% | 10003 |
| 0.5-0.6 | 0.84% | 3768 |
| bellow 0.5 | 0.7% | 3290 |

(b)

Out of a total of 762782 predicted boxes in the inpainted image set, we found that only 4455 boxes (about 0.59%) could not be mapped to the boxes detected in the original image with the same class and an overlap of at least 0.5 IoU. In comparison, 726969 boxes (96.8%) in the inpainted images have a matching box with the same class and a nearly identical spatial location (IoU > 0.7) in the original image. Moreover, in Figure 5, we also provide visual examples of the resulting images after the inpainting process. As can be seen, the inpainting images are highly similar to the original ones. The editions are visually indistinguishable even when large objects are removed. Despite the inpainting strategy being relatively under-explored, we conclude that the observed performance gaps in MERLIM should not be attributed to the minimal discrepancies identified by representative image networks (ResNet50, YOLOv7, Mask R-CNN) but rather to hallucination events in the IT-LVLM.

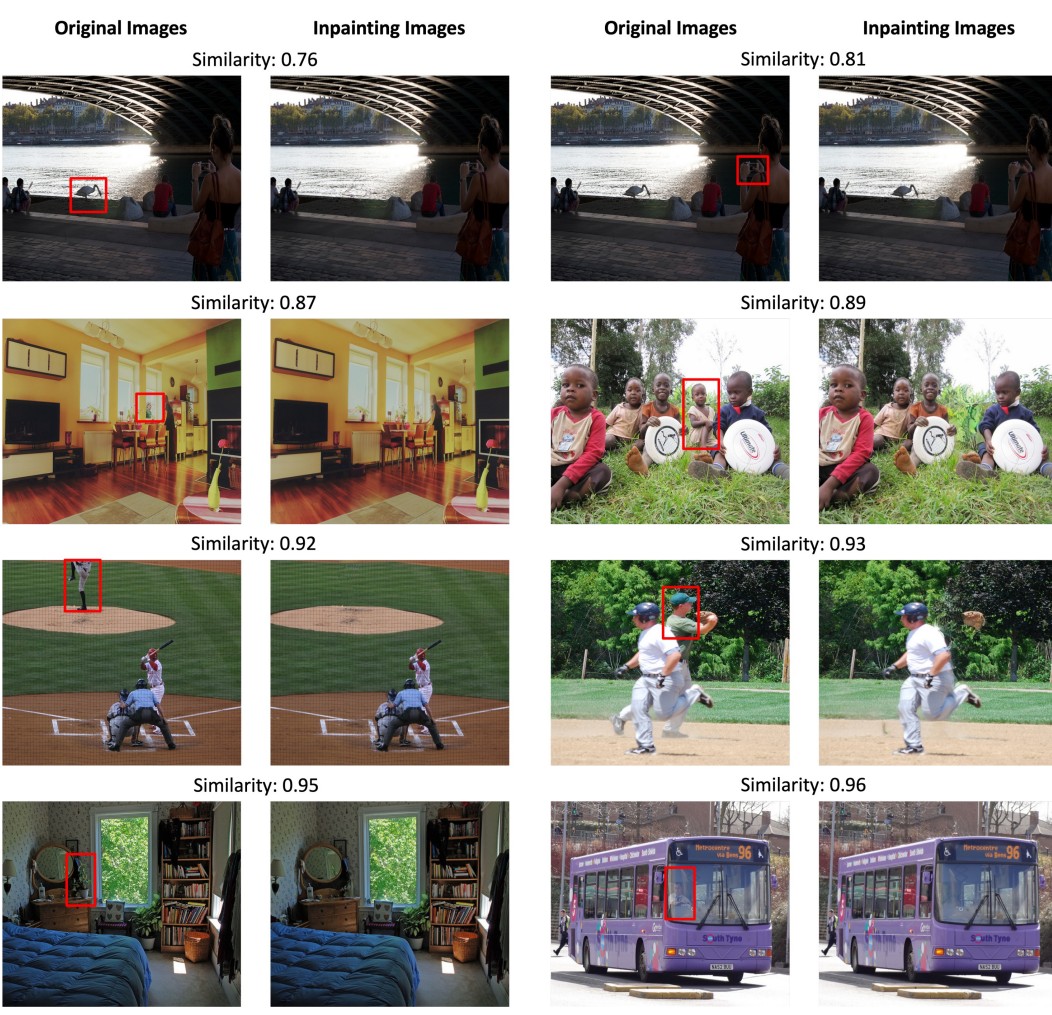

Figure 5: **Examples of Inpainting images.** We present eight examples demonstrating the results of removing an object from the original image using the inpainting model proposed by Li et al. (2022b). As illustrated, the edits are visually seamless, even when large objects are removed, making them nearly indistinguishable from the original.

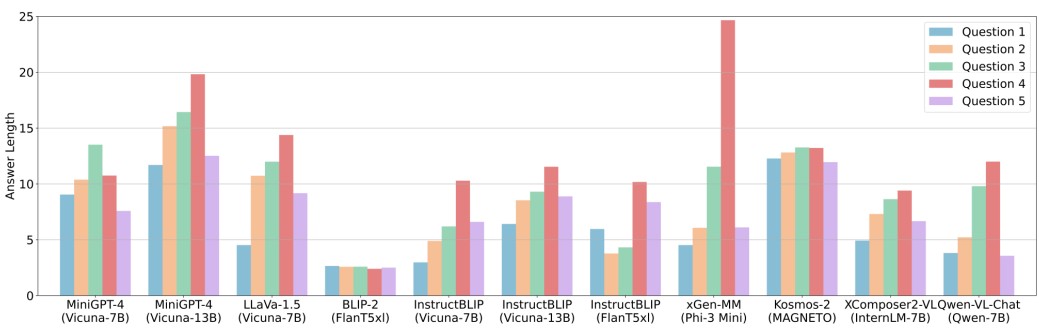

Figure 6: **Number of nouns predicted.** We extract the nouns using the spaCy library from the answers of the models from all the formulated prompts on the original image set. It is worth noting that BLIP2 predicts a consistent number of nouns across all the prompts, unlike the other methods.

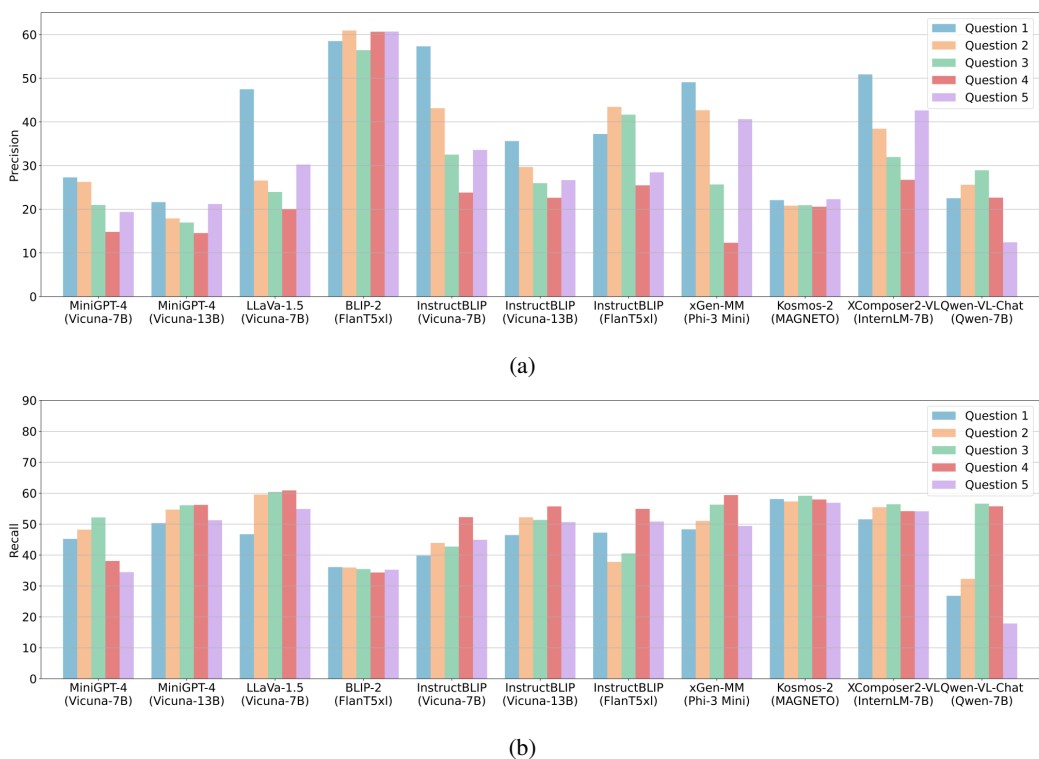

(a)

(b)

Figure 7: **Precision and Recall on the original image set.**. We compare the Precision and the Recall of IT-LVLMs on the original image set across the five proposed prompts $P$ in the sub-figure **a)** and **b)**, respectively. It is worth noting that the biggest models, such as InstructBLIP and MiniGPT4 with Vicuna13B and LlaVa, get the highest recall and lowest precision.

## A.2 OBJECT HALLUCINATION

In Figure 6, we provide further evidence of instruction bias in IT-LVLMs. Despite the semantic equivalence of the proposed prompts, all methods, except for BLIP-2 Li et al. (2023a) and Kosmos-2 Peng et al. (2024), generate varying numbers of nouns for each prompt. Figure 7 illustrates that while BLIP-2 produces the shortest answers (in terms of noun count), it consistently achieves higher precision compared to other methods, albeit with lower recall. This indicates that BLIP-2 is less prone to hallucination, likely because it was trained to generate concise captions rather than detailed descriptions. On the other hand, larger models trained with instructional data, such as InstructBLIP and MiniGPT-4 with Vicuna-13B Dai et al. (2023); Zhu et al. (2023), LLaVa Liu et al. (2023), and xGen-MM Xue et al. (2024), tend to generate longer answers on average. This increases the likelihood of hallucinations, particularly when visual grounding is insufficient, resulting in higher recall but lower precision compared to BLIP-2. As shown in Figure 8 shows that even a proprietary model such as GPT-4o-mini OpenAI (2023) presents a high hallucination rate.

## A.3 LLMs vs IT-LVLMs

In Table 4, we compare the performance of IT-LVLMs to their corresponding LLMs without visual input. Since the LLMs lack the capability to process visual information, we allow them to answer "do not know." Using the original questions from the visual relationship task (denoted as **Question**), we prompt an LLM as follows: "$p_{llm}$ = **Question**. *Please answer yes, no, or do not know*". While LLMs sometimes respond with "I do not know", they typically base their answers on the knowledge acquired during language pre-training. With three response options, the random chance of selecting the correct answer is 33%.

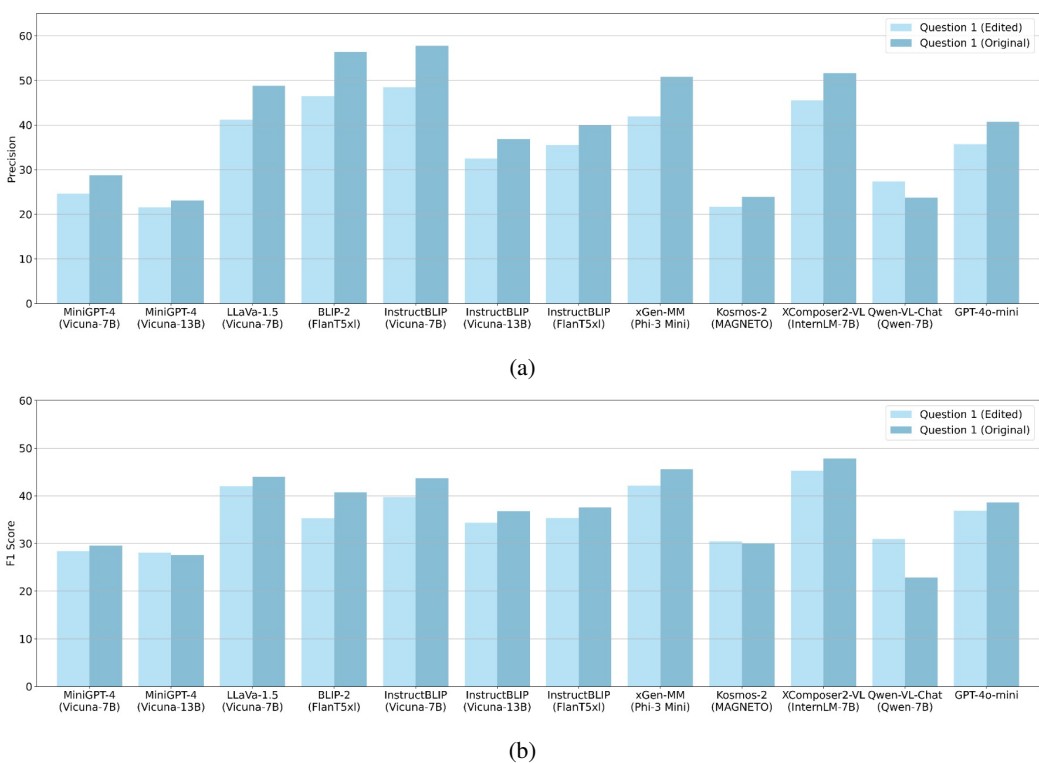

(a)

(b)

Figure 8: **Comparing GPT-4o-mini with the open-source models on the original and edited sets.**
(a) We compare the precision of IT-LVLMs on the edited and original image sets using one prompt.
To analyze the hidden hallucination problem, we focus on the subset where the image inpainting
removes an entire category. Notably, all methods lose performance on the edited image set. (b) We
also report the F1 Score on the original and edited images using the same question.

Table 4: **Textual Biases.** We compare the accuracy of IT-LVLMs in the relationship understanding
with the performance of LLMs. Since the LLMs lack visual context we allow the model to reply
'yes', 'no' or 'I Don't know'. We report two accuracy values, one where we penalized every 'I don't
know' as a false prediction (left side of /) and the other considering only the questions answered with
yes or no (right side of /). FlanT5xl outperforms the other opensource LLMs.

| Model | LLM | Random Set | | Curated Set | |
|---|---|---|---|---|---|
| | | $\text{Acc}_{org} \uparrow$ | $\text{Acc}_{org}^{neg} \uparrow$ | $\text{Acc}_{org} \uparrow$ | $\text{Acc}_{org}^{neg} \uparrow$ |
| Random Baseline | N/A | 33% / 50% | 33% / 50% | 33% / 50% | 33% / 50% |
| LLM Only | ChatGPT 3.5 | 21.27% / 80.01% | 8.05% / 28.83% | 13.06% / 55.80% | 11.57% / 47.78% |
| LLM Only | Vicuna-7B v1.1 | 6.59% / 44.72% | 27.53% / 66.86% | 11.11% / 43.05% | 26.42% / 55.54% |
| LLM Only | Vicuna-13B v1.1 | 7.92% / 75.22% | 2.07% / 25.23% | 7.63% / 55.60% | 4.92% / 47.76% |
| LLM Only | FlanT5xl | **41.82%** / 69.47% | **40.06%** / 55.84% | **36.10%** / 54.65% | **50.74%** / 65.80% |
| BLIP-2 | FlanT5xl | 83.62% | 48.80% | 67.84% | 49.77% |
| InstructBLIP | Vicuna-7B v1.1 | **91.17%** | 19.60% | **76.75%** | 45.31% |
| InstructBLIP | Vicuna-13B v1.1 | 90.32% | 16.24% | 75.99% | 43.52% |
| InstructBLIP | FlanT5xl | 80.69% | **77.16%** | 70.15% | **63.90%** |

For additional evaluation (shown on the right side of the table), we consider only the answers where
the LLM responded with either "yes" or "no." Although Vicuna performs below the random baseline,
its performance improves significantly when focusing solely on yes/no responses, especially for
Vicuna-13B. FlanT5xl exceeds random chance and further narrows the performance gap between
LLMs and IT-LVLMs.

We conclude that the language model serves as a strong prior for resolving visual relations, even
without image information. These textual priors can act as "shortcuts" in MERLIM, contributing

to the performance difference between the Random and Curated sets, where these biases are less effective. While these textual "shortcuts" can occasionally align with visual grounding (Hidden Hallucinations), they are often revealed when visual grounding contradicts language-based intuitions.

## A.4 GRADIENT ANALYSIS

To further investigate the visual grounding of IT-LVLMs, we take inspiration from Selvaraju et al. (2017) and compute the proportion of the total gradient attributable to each specific type of input (Image, Question, Answer) for each predicted token. We analyze the model's output logits before token selection and propagate the gradient to the input by identifying the maximum activation value per output token. Specifically, for each output token, we determine the gradient contribution from each input type and then average these contributions across all answers. To simplify our analysis, we focus on architectures with autoregressive LLMs, such as Vicuna, which explicitly use previous output tokens as inputs for predicting subsequent tokens.

As shown in Figure 5, the gradient of the visual input is significantly lower than that of the language inputs (Question and Answer). This indicates that LLMs prioritize language tokens over visual ones when predicting answers, leading to hallucinations based on language biases. Additionally, we observe that only a few tokens account for most of the visual gradients. For instance, in InstructBLIP with Vicuna-7B, just 10 out of 30 tokens represent nearly 73% of the visual gradients, making it challenging for specific and strong visual information to be adequately considered. These findings support the results of our previous experiments.

## A.5 NUMBER OF UNSUITABLE OUTPUTS

On the Object Recognition task models will occasionally provide responses lacking any valid nouns, such as "There are many objects" or "It is a beautiful scene". In such cases, we set both recall and precision metrics to 0. The results in Table 6 outline that MiniGPT-4 with Vicuna-7B v0 and BLIP-2 with FlanT5xl produce the largest number of unsuitable answers. Conversely, the InstructBLIP methods consistently generate valid responses with object lists.

## A.6 VISUAL EXAMPLES OF MERLIM'S TASKS.

Figure 10 presents additional visualizations of the tasks evaluated by MERLIM. It shows the InstructBLIP outputs for both the original and edited images across the three tasks. Notably, the model consistently produces visually ungrounded predictions (hallucinations) in all scenarios, highlighting the complexity of the hallucination problem in these instructional models.

## A.7 IMPACT STATEMENTS

MERLIM is a benchmark designed to assess the performance of Instruction Tuning Large Vision and Language Models (IT-LVLMs). Besides the empirical evaluation, MERLIM's primary aim is to identify and quantify instances of "Hallucination" events in the textual responses generated by IT-LVLMS. Consequently, MERLIM represents a tool with a potentially positive societal impact by encouraging advanced IT-LVLM models to be more robust to hallucination events and, therefore, bring more factual and informative responses.

Table 5: **Study of the relevance of the inputs.** We analyze the relevance of tokens from each kind of input (Image, Question, and Answer tokens) by computing the portion of the total gradient produced in the input that belongs to the image, question, and answer tokens relative to the outputs. Specifically, for each output, we calculate the gradient contribution from each input type and then average these contributions across all answers.

| Model | Vis. Tokens | Que. Tokens | $\nabla_{\mathbf{vis}} \uparrow$ | $\nabla_{\mathbf{que}} \uparrow$ | $\nabla_{\mathbf{ans}} \uparrow$ |
|---|---|---|---|---|---|
| LLaVA-1.5 | 575 (All) | 52 (All) | 27.68% | 44.75% | 28.67% |
| LLaVA-1.5 | top 10 | top 10 | 11.47% | 53.28% | 36.68% |
| InstructBLIP | 32 (All) | 10 (All) | 24.89% | 47.45% | 29.04% |
| InstructBLIP | top 10 | top 10 | 19.98% | 52.01% | 29.41% |

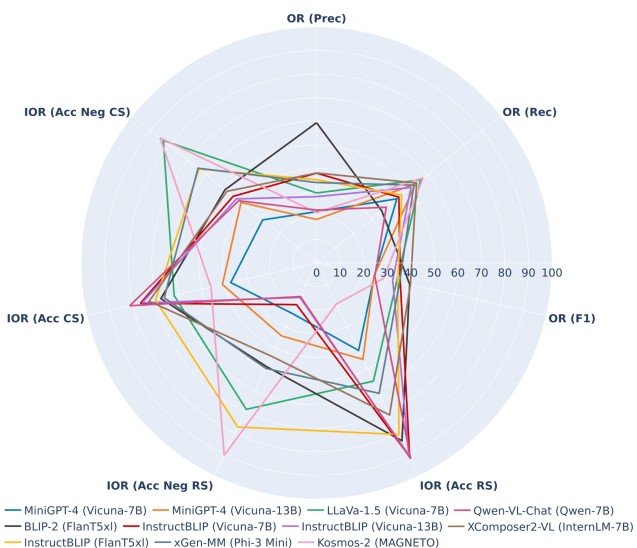

Figure 9: **Results on MERLIM.** We showcase a subset of the performance evaluations on MERLIM. Our evaluation metrics include Precision (Prec), Recall (Rec), and F1 Score (F1) for the Object Recognition Task (OR). We also measure the Accuracy at identifying inter-object relationships (IOR) in two sets: one set generates negative examples through random sampling (Random - Sample RS), and the second set has curated relations (Curated Set - CS) where a commercial LLM discards impossible associations, thus forcing the IT-LVLM to use the visual data. We further verify the IT-LVLMs consistency, as we calculate the Accuracy for both affirmative (Acc) and negative (Acc Neg) versions of the instructions describing the object relations.

Table 6: **Number of unsuitable outputs.** The IT-LVLMs occasionally provide unsuitable answers to the five proposed questions for the Object Recognition Task. That is answers without valid nouns, for instance, *"There are many objects"* or *"It is a beautiful scene"*. For such instances, we establish both recall and precision metrics as 0.

| Model | LLM | Num. unsuitable outputs | | | | | | | | | |
|-------|-----|------------|------|------------|------|------------|------|------------|------|------------|------|
| | | Question 1 | | Question 2 | | Question 3 | | Question 4 | | Question 5 | |
| | | Original | Edited | Original | Edited | Original | Edited | Original | Edited | Original | Edited |
| MiniGPT-4 | Vicuna-7B v0 | 17 | 110 | 6 | 28 | 2 | 26 | 10 | 61 | 81 | 414 |
| MiniGPT-4 | Vicuna-13B v0 | 0 | 2 | 0 | 0 | 0 | 0 | 1 | 10 | 0 | 1 |
| LLaVA-1.5 | Vicuna-7B v1.5 | 0 | 1 | 0 | 0 | 0 | 0 | 0 | 0 | 0 | 0 |
| BLIP-2 | FlanT5xl | 11 | 73 | 6 | 58 | 5 | 65 | 21 | 209 | 15 | 96 |
| InstructBLIP | Vicuna-7B v1.1 | 0 | 1 | 0 | 2 | 0 | 0 | 0 | 0 | 0 | 0 |
| InstructBLIP | Vicuna-13B v1.1 | 0 | 2 | 0 | 0 | 0 | 0 | 0 | 0 | 0 | 0 |
| InstructBLIP | FlanT5xl | 0 | 0 | 0 | 1 | 0 | 1 | 0 | 0 | 0 | 0 |
| xGen-MM | Phi-3 Mini 3.8B | 0 | 0 | 0 | 0 | 0 | 0 | 0 | 0 | 0 | 0 |
| Kosmos-2 | MAGNETO | 0 | 0 | 0 | 0 | 1 | 12 | 0 | 0 | 0 | 0 |
| XComposer2-VL | InternLM-7B | 4 | 36 | 12 | 180 | 0 | 1 | 0 | 1 | 4 | 2 |
| Qwen-VL-Chat | Qwen-7B | 6 | 38 | 2 | 4 | 0 | 1 | 0 | 4 | 3 | 22 |

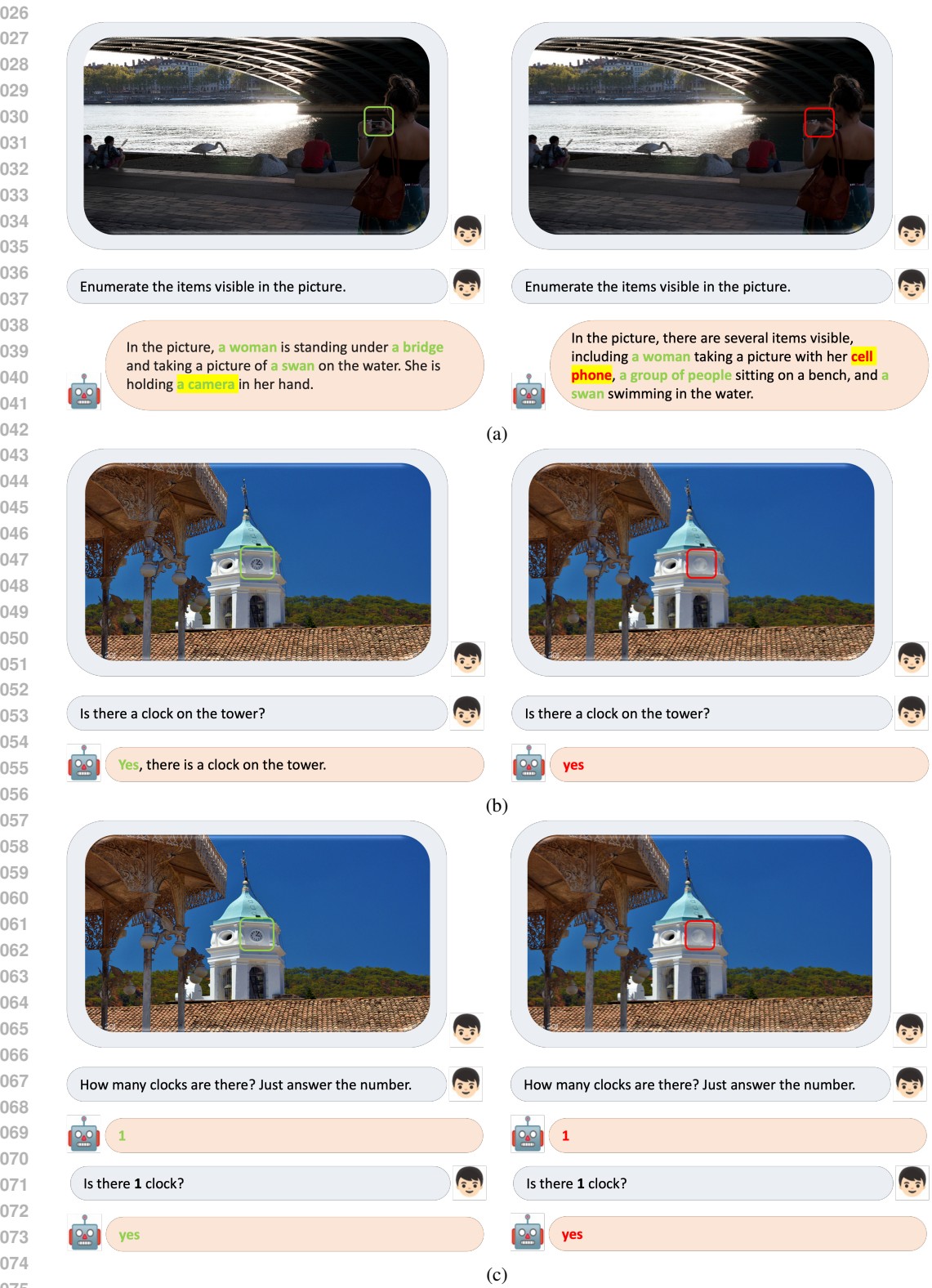

Figure 10: **Visual Examples.** We present additional visual examples illustrating the tasks evaluated by MERLIM. Sub-figures **(a)**, **(b)**, and **(c)** showcase comparisons of InstructBLIP outputs for original and edited images, focusing on the tasks of Object Recognition, Inter-Object Relationship Understanding, and Object Counting, respectively.

