# OpenReview forum: "Behind the Magic, MERLIM: Multi-modal Evaluation Benchmark for Large Image-Language Models"
_ICLR.cc/2025/Conference — Submitted to ICLR 2025_

### Official Review · Reviewer_YPpW · 2024-11-02

**Soundness:** 2
**Presentation:** 2
**Contribution:** 2
**Rating:** 3
**Confidence:** 5

**Summary:**

The paper introduces MERLIM, a multi-modal evaluation benchmark for assessing Instruction Tuning Large Vision and Language Models (IT-LVLMs) on fundamental computer vision tasks. While the paper aims to provide a scalable test-bed to evaluate IT-LVLMs, it relies heavily on reusing and editing images from the MS-COCO dataset, which significantly diminishes the novelty and contribution of the work as a new benchmark. The paper also fails to incorporate evaluations of newer models that have emerged in the past year, limiting its relevance and timeliness. Additionally, the lack of open-sourcing the benchmark data hinders its practical utility and community adoption.

**Strengths:**

- The paper try to address a pertinent issue in the field of MLLMs, which is the need for a standardized evaluation benchmark to assess their performance on computer vision tasks.

- The focus on identifying hallucination events in MLLMs is a valuable contribution, as it highlights a critical challenge in the reliability of these models.

- The paper provides a detailed analysis of several MLLMs, shedding light on their limitations and potential areas for enhancement.

**Weaknesses:**

- The paper's reuse of MS-COCO images and ground-truth annotations as the primary dataset for the benchmark is a significant concern.  This approach potentially introduces biases and data leakage, which can affect the evaluation's fairness and accuracy.  A new benchmark should ideally utilize a fresh dataset to provide an unbiased assessment of model performance.

- The paper does not include evaluations of newer MLLMs that have emerged since the initial submission, which is a critical oversight given the rapid advancements in the field.  This limits the benchmark's relevance and timeliness.

- The lack of open-sourcing the benchmark data is a substantial drawback.  Open-sourcing is a standard practice that fosters transparency, reproducibility, and collaborative improvement.  Benchmarks like POPE, NExT-GQA, and HallusionBench have demonstrated the value of open-sourcing by enabling broader community engagement.

**Questions:**

- Regarding the lack of open-sourcing the benchmark data, could the authors explain the rationale behind this decision?


- How does MERLIM compare to other existing benchmarks in terms of its ability to detect and quantify hallucination events?     Could the authors provide a comparative analysis to highlight the unique aspects of MERLIM?

---

> ### Author Response · Authors · 2024-11-24
>
> We thank reviewer YPpW for their comments and questions. Below, we provide detailed clarifications and responses to each of the questions.
>
> **W1 MS-COCO Bias.** Similar to MERLIM, POPE and NExT-GQA are based on a single dataset (MS-COCO and NExT-QA), HallusionBench contains only 346 images, many of them are 2D plots and visual illusions, with very few natural images. In comparison, MERLIM uses the full validation set of MS-COCO (POPE uses only 500 images) and generates new data (changes are subtle, but the images are never semantically identical). Due to these changes, MERLIM can break some standard visual biases (for example, in Figure 1, there are no visible tennis racquets on a tennis court). Moreover, we continue with such a visual edition procedure and evaluate its effects in 3 different tasks.
> Therefore, we politely disagree with YPpW and state that we consider MERLIM as a direct tool to break some of the standard visual biases and analyze the effects of this visual disentanglement in multi-modal tasks.
> Finally, we invite YPpW to reconsider its requirement for a fully unbiased dataset. Computer vision datasets (and, by extension, the models trained on them) are highly biased despite modern techniques that collect large-scale data from diverse sources. Current (and classic) datasets suffer from strong bias to the point that tuned models can easily identify the source dataset by simply analyzing one image [A].
>
> [A] Liu, Z., & He, K. (2024). A Decade's Battle on Dataset Bias: Are We There Yet?. arXiv preprint arXiv:2403.08632.
>
> **W1 Leakage.** We welcome YPpW observation in leakage, as this is a common issue in evaluating LLMs. However, we politely disagree as MERLIM does not have a higher risk of leakage than any other benchmark. It is impossible to fully guarantee that MERLIM images will not be used as training data. We share YPpW enthusiasm about open data, thus we have already made available the entirety of  MERLIM data (labels included). Even if we created a new  test-set with no ground-truth data, those new images must be downloaded for inference, then they can be manually labeled and used for training.
> We further clarify that MERLIM uses only COCO VALIDATION data, and the 31K edited images correspond to altered versions of MS-COCO. A significant data leakage can only occur if someone directly trains on the validation data of COCO along with the derived data of MERLIM.
> In summary, MERLIM does not pose a greater (or smaller) risk of leakage than any other dataset new or old. leakage will only occur if an actor (by omission or malice) decides to train our data, which is the case for any other benchmark regardless of its recency.
>
> **W2 Laking of evaluations of newer MLLMs.** The field of LVLMs is advancing rapidly. This makes it challenging to evaluate every newly developed IT-LVLM. Nevertheless, we assessed 11 state-of-the-art models to demonstrate the advantages of our benchmark, MERLIM, in highlighting the visual grounding limitations and language biases of IT-LVLMs. We plan to continuously update the leaderboard to include newly released state-of-the-art models, ensuring MERLIM remains a relevant and comprehensive evaluation framework.
>
> **W3. Lack of open-sourcing of the benchmark data.** Similar to POPE, NExT-GQA, and HallusionBench, our data is open and free of charge to whoever wants to use it. Our license is the very same as MS-COCO “Creative Commons Attribution 4.0 License”. The original MS-COCO data can be downloaded from the official website.
>
> We made the MERLIM’s inpainted data, queries and code available at submission time (anonymous repository). The link to the data (inpainted images and queries) can be found at the end of the introduction (line 101) on the word ‘here’. The format does not show as a standard hyperlink but can be clicked. We have corrected the style of this link in the revisited version.
>
> For completeness, we have updated our anonymous repository to explicitly include the license of our inpainted data and related queries.

---

> ### Author Response · Authors · 2024-11-24
>
> **Q1 Open Data.** Please refer to the answer for W3. Lack of open-sourcing of the benchmark data.
>
> **Q2 Comparison to other benchmarks.** As highlighted by reviewer 4cSE, the rigorous analysis in MERLIM discovers ‘Hidden Hallucinations’, a phenomenon where seemingly correct answers lack any visual grounding and therefore constitute an error. To the best of our knowledge, MERLIM is the only tool that can identify and quantify such a source of error.
> As stated by dTrJ, MERLIM works with the open-set natural language generated by the language decoder. This allows for a more realistic assessment of the capabilities of the IT-LVLMs.
> As outlined by 1AUQ, MERLIM focuses on fundamental visual tasks, which are overlooked by other benchmarks. Critically, we identify that despite the large-scale pre-training and all the in-domain returning IT-LVLMs, they still underperform in these tasks. We hypothesize that more complex tasks, such as captioning and VQA, could be affected by these fundamental errors in identifying and locating objects in an image.
> Unlike others, MERLIM constitutes a large-scale Benchmark with 2 orders of magnitude more data than POPE, HallusionBench, and NExT-QA. Moreover, MERLIM’s clever way of editing visual data allows it to disentangle some of the most common regularities in visual data.
> We conclude that given its scale, evaluation strategy, and visual manipulation strategy, we see MERLIM as a more real-world evaluation benchmark than the currently available benchmarks.

---

> > ### Comment · Reviewer_YPpW · 2024-11-26
> >
> > 1. The author's response does not address my concerns. In my view, building a new benchmark on top of an old one is a lazy approach. Additionally, I disagree with the practice of rigidly adhering to the traditional computer vision benchmark, COCO, in the multimodal field. While many hallucination benchmarks based on COCO have been accepted, that is not a justification for accepting this paper. There are already too many hallucination benchmarks based on COCO, which is detrimental to the development of the field. As a benchmark study, failing to collect new real-world data severely undermines its contribution.
> >
> > 2. If an MLLM was trained using the COCO train set, it might achieve better scores on the benchmark proposed in this paper. Conversely, an MLLM that did not use the COCO train set might score poorly. However, this does not reflect the true hallucination level of the model, i.e., its real-world hallucination performance, but rather its hallucination performance within the COCO domain. Such an evaluation is biased and does not facilitate an accurate assessment of the capabilities of current MLLMs.
> >
> > 3. The author did not evaluate recent closed-source models, such as GPT-4V, GPT-4o, or Claude, nor did they assess newer MLLMs like LLaVA-Next or LLaVA-OneVision. In my opinion, this lack of effort makes the benchmark less convincing.

---

> ### Author Response · Authors · 2024-11-27
> **Official Comment by Authors [1/2]**
>
> We thank Reviewer YPpW for engaging in the discussion and for his/her comments. Regarding YPpW assertion that *“The author's response does not address my concerns”* In the original review, YPpW refers to weakness regarding the lack of open-source data and potential data leakage in MERLIM, highlighting the first one as “ a substantial drawback”. We note that after our rebuttal, these weaknesses are no longer discussed by YPpW.
> If concerns about the license of the data or its potential leakage remain, we invite YPpW to address them directly in this discussion. We would be more than happy to provide additional feedback and clarification. Otherwise, we politely invite YPpW to acknowledge that these concerns have been addressed and revisit his/her rating, given that the only drawback that was highlighted as “substantial" has been addressed.
>
> Regarding the remaining concerns. We proceed item by item over YPpW ‘s last message.
>
> 1. *"While many hallucination benchmarks based on COCO have been accepted, that is not a justification for accepting this paper. There are already too many hallucination benchmarks based on COCO.”*
>
> We politely disagree with YPpW and invite the reviewer to list other hallucination benchmarks that evaluate instructional models (which are distinct from visual-language models). Moreover, we invite YPpW to carefully analyze the performance of modern IT-LVLMs in such benchmarks, POPE and MERLIM are far from being saturated. Certainly MS-COCO is a classic computer vision dataset, but the recent empirical evidence indicates that it remains a challenging test-bed for current IT-LVLMs.
>
> 2. *“Rigidly adhering COCO in the multimodal field”.*
>
> Our benchmark does not represent the standard practice of object detection/recognition in MS-COCO. Instead, MERLIM provides a more real scenario than the existing hallucination benchmarks. We evaluate the hallucination on open-ended questions (not part of the MS-COCO dataset) and provide a set of questions to analyze the language bias (no language biases are assessed in the MS-COCO dataset). Finally, the LLM of the instructional model can output open answers, we also consider if the predicted nouns are synonyms of the ground truth objects, this is not part of the MS-COCO dataset or its evaluation procedure.
>
> 3. *“If an MLLM was trained using the COCO train set, it might achieve better scores on the benchmark proposed in this paper”.*
>
> We invite the reviewer to YPpW to revisit figures 2 and 3. xGen-MM (BLIP-3) does not rely on MS-COCO during training but outperforms models such as BLIP2, InstructBLIP, and LLaVA-1.5, which are actually trained on MS-COCO data. This empirical evidence is in direct contraposition to YPpW assertion about dataset biases. We remind YPpW that MS-COCO is just one of several datasets used in the training of IT-LVLMs and that the training typically comprises a visual-language alignment step followed by an instructional tuning step. None of these steps directly includes the task of object recognition, and these training steps only use MS-COCO training data.
> Our empirical results suggest that the difference in the training objectives of IT-LVLMs and MERLIM’s primary tasks does not generate a clear overfit or constitute any evident data leakage in the MS-COCO dataset.
>
> 4. *“However, this does not reflect the true hallucination level of the model, i.e., its real-world hallucination performance, but rather its hallucination performance within the COCO domain.”*
>
> We politely invite YPpW to clarify what he/she calls “real-world hallucination performance”, specifically, how can the hallucination rate of a model be estimated independently of the benchmark data?. Moreover, we also invite YPpW to elaborate on what kind of dataset can effectively evaluate real-world hallucinations regardless of the domain of the data. We will proceed to directly compare the most relevant design choices of MERLIM with the outlined requirements of YPpW.
> We must note that collecting new data (as suggested by YPpW) would also receive the YPpW complaint. YPpW argues that COCO data can only assess hallucinations in the COCO domain. Likewise, the newly collected dataset will only assess hallucinations in its collection domain.
>
> Finally, we would like to highlight that MS-COCO was “collected images from Flickr, which tends to have fewer iconic images. Flickr contains photos uploaded by amateur photographers with searchable metadata and keywords”(Lin et al., 2014),. Thereby reducing potential biases in collection and labeling. Furthermore, we consider the different open-ended questions would mimic a closer to the real-world working scenario.

---

> ### Author Response · Authors · 2024-11-27
> **Official Comment by Authors [2/2]**
>
> *State-of-the-Art Models*
>
> Our benchmark evaluates 11 models, including the recent xGen-MM (BLIP-3), which was first submitted to arXiv on August 16, 2024. We recognize that this is a rapidly evolving field, making it challenging to evaluate every newly released model. Nevertheless, our benchmark provides a comprehensive analysis of state-of-the-art models, offering valuable insights that can inspire future research. Additionally, in our updated revision, we have incorporated evaluation results for GPT-4o-mini, now included in Figure 8 of the supplementary material.

---

> > ### Comment · Reviewer_YPpW · 2024-11-27
> >
> > I believe the biggest flaw of this work is not the potential data leakage, but the fact that it is entirely built upon COCO.  As the authors mentioned, the COCO dataset was collected from the Flickr domain prior to a specific year (2014), which inherently introduces a strong bias.  These data are outdated and inevitably overlap with newer datasets or training sets developed after 2014, creating potential risks of data leakage.
> >
> > For a benchmark work in 2024, relying on old data rather than collecting fresh datasets demonstrates a lack of contribution to the field.  Additionally, the range of models evaluated by the authors is neither rich nor extensive.  Evaluating more open-source models is not a particularly difficult task, yet the authors appear unwilling to do so.
> >
> > Therefore, my concerns remain entirely unresolved.  I believe this paper does not meet the standards for acceptance, and I will maintain my rating of 3.

---

> > > ### Author Response · Authors · 2024-11-27
> > >
> > > We thank reviewer YPpW for the prompt response and the engagement in the discussion thread of this paper. While we welcome YPpW enthusiasm for our submission, we strongly disagree with his/her last assessment, where the primary reject argument is the leakage risks of data in COCO, given the year of the data collection.
> > >
> > > In addition to our previous arguments, we highlight that many widely used datasets have a similar collection date (even earlier) and remain fundamental to computer vision research. For instance, ImageNet (introduced in 2009) is commonly used for pretraining in the image domain. Something-Something (introduced in 2017) and Kinetics (introduced in 2017) are widely employed in training and evaluation in the video domain. Regarding visual questions answering benchmarks, VQA (introduced in 2015) and SQA (introduced in 2016) retain their relevance. Recently, their value has been further recognized as they have become significant benchmarks to evaluate the multi-modal capabilities of  IT-LVLMs.
> > > Despite their age, these datasets (and several others) continue to serve as standard tools for benchmarking and pre-training tasks in computer vision, indicating that the collection date alone is not the reason why the scientific community discards or accepts datasets.
> > >
> > > To further complement this discussion, we invite reviewer YPpW to provide references that support his/her claim that a dataset or benchmark must be directly discarded if the collection date of its images has passed a certain timestamp.

---

> > > > ### Comment · Reviewer_YPpW · 2024-11-28
> > > >
> > > > I don't think a dataset or benchmark should be dismissed solely because its image collection date surpasses a certain timestamp.    However, this does reduce the contribution of this work.    This benchmark heavily relies on existing ones, specifically COCO, and the lack of freshness in its dataset is undeniable.    Additionally, it does not introduce significant new data to the community.    As a developer of MLLMs, from the perspective of developing and evaluating these models, I am confident this benchmark's contribution will be limited.    I do not think that adding another benchmark based on COCO offers meaningful value to model evaluation.  Consequently, I am raising my confidence score to 5.

---

> > > > > ### Author Response · Authors · 2024-11-28
> > > > >
> > > > > We thank YPpW for the continuous engagement in this discussion and unwavering commitment to our submission. We would like to remind YPpW that the review process should not include personal opinions. Instead, every claim (by authors and reviewers) should be justified with references or empirical results.
> > > > >
> > > > > We politely disagree with the YPpW’s claim *“I don't think a dataset or benchmark should be dismissed solely because its image collection date surpasses a certain timestamp. However, this does reduce the contribution of this work”*. As we have already outlined in this discussion, there are many datasets that remain significant and impactful despite their release date (MS-COCO is one them). We must emphasize that YPpW has yet to provide any evidence (or references) that indicate that leakage over MS-COCO has already surpassed a threshold that makes our benchmark unacceptable. Neither has YPpW provided a substantiated argument on why a benchmark must include unreleased data to recommend acceptance.
> > > > > We remind YPpW that the confidence rating of a reviewer reflects the expertise of said reviewer with the topic of a submission. We simply can not understand how our last comment improved YPpW’s knowledge and overall experience in the domain of IT-LVLMs and hallucinations.
> > > > >
> > > > > Finally, we would kindly request YPpW to abstain from using fallacious arguments. In an academic discussion the fact that YPpW has a certain role/title does not support or validate his/her arguments, nor invalidates our arguments. In short, we invite YPpW to revisit his/her proceedings during this discussion and provide scientific evidence (not personal statements) to support his/her argument or to refute ours.

---

### Official Review · Reviewer_4cSE · 2024-11-02

**Soundness:** 3
**Presentation:** 2
**Contribution:** 3
**Rating:** 5
**Confidence:** 3

**Summary:**

This paper introduces MERLIM, a novel multi-modal Benchmark for evaluating IT-LVLMs for hidden hallucinations through several basic computer vision tasks.

**Strengths:**

This paper propose the concept of hidden hallucinations in LVLM, a phenomenon  that has been overlooked in the community. This brings insights on the performance of state-of-the-art IT-LVMLs including limitations at identifying fine-grained visual concepts.

**Weaknesses:**

1. The writing of this article needs to be strengthened. Figure 1 highlights the finding of "hidden hallucinations", but this is not elaborated in detail in the abstract. I was wondering if "hidden hallucinations" is the key point of this paper?
2. The focus of the article is not clear. The core of the article seems to be the discovery of the phenomenon of hidden hallucinations. However, in the experiments, too much emphasis appears to have been placed on the mention of fundamental visual tasks. Nevertheless, there are already benchmarks for evaluating fundamental visual tasks nowadays.

**Questions:**

1. The article seems to have two focal points, one is to conduct evaluations on fundamental computer vision problems, and the other is on hidden hallucinations. Which is the focus of this paper? From my perspective, you are trying to demonstrate the language bias and limitation of vision groundings in LVLM from the phenomenon called "hidden hallucinations" through several computer vision problems (Object Recognition, Object Counting, and Inter-object Relationship Understanding), is my understanding right?
2. What is the difference between IT-LVLM and LVLM? Why does the article emphasize IT-LVLM instead of directly using the term LVLM which is widely used?

---

> ### Author Response · Authors · 2024-11-24
>
> We thank reviewer 4cSE for the insightful review. We will take care of introducing hidden hallucinations right from the abstract. As highlighted by 4cSE, this is a central contribution of our benchmark and an under-explored topic in the research community.
>
> **W2, Q1. Focus of the article.** Indeed, the "hidden hallucinations" are a direct contribution of this paper, but our contribution is not limited to them. Many hallucination benchmarks follow the style of VQA by querying the IT–LVLM with multiple-choice answers. These benchmarks do not control for alternative questions with similar semantics or allow for open-set responses. Since MERLIM design includes these two aspects, we see our benchmark as a more realistic scenario for IT–LVLMs. These models are commonly used as multi-modal assistants that reply with natural language.
>
> In addition, MERLIM is the first benchmark to employ subtle visual edits, thus establishing a connection between effective visual grounding and IT–LVLM performance. We think MERLIM can bring valuable insights.
>
> Finally, as outlined by 4cSE, we also focus on fundamental vision tasks. As most benchmarks follow the VQA approach, the community has overlooked these tasks. MERLIM shows that, despite building on large visual encoders pre-trained with large-scale data, current IT–LVLMs underperform in fundamental vision tasks. In fact, MERLIM shows that none of them can reach a 50% F1 score in object recognition.
>
> **Q2. Difference between IT-LVLM and LVLM?** The key difference between these modes is the instructional training and the capacity of the language module. CLIP would be an LVML, it lacks any instructional training and has a relatively smaller language module. Therefore, it can only align multi-modal data. Meanwhile, IT–LVLMs like LLaVA, BLIP-2, and Kosmos-2 build upon a stronger language component and can output language with their decoder component. After instructional training, the IT–LVLMs can follow natural language instructions and produce open-set responses according to the language instruction and the image data.

---

### Official Review · Reviewer_Qhva · 2024-11-04

**Soundness:** 3
**Presentation:** 3
**Contribution:** 3
**Rating:** 6
**Confidence:** 3

**Summary:**

This paper presents a new evaluation benchmark for Instruction Tuning Large Vision and Language models (IT-LVLMs), dubbed MERLIM, containing over 300K image-question pairs. This benchmark focused on three types of tasks: Object Recognition, Object Counting, and Relationship Understanding. Besides, this benchmark enables to identify two types of hallucinations: Regular Hallucinations and Hidden Hallucinations, the latter of which is newly introduced in this paper. In experiments, the authors have evaluated many IT-LVLMs, obtaining several insights on the performance of state-of-the-art IT-LVMLs.

**Strengths:**

- The newly introduced evaluation aspect *Hidden Hallucinations* can help better understand the Hallucination issue of IT-LVLM.
- The analyses on the performance of state-of-the-art IT-LVMLs are beneficial.
- This paper is clearly-written, with a well-structured presentation of MERLIM, including its detailed design and evaluation methodology.

**Weaknesses:**

- The included tasks such as Object Recognition and Object Counting seem to have overlaps with previous benchmarks [1]. It would be better if the authors could include detailed comparisons between the proposed benchmark and existing ones. It may be helpful to more thoroughly analyze the dataset.
- The field of LVLMs is constantly developing. Many models evaluated in the paper may not be that strong nowadays. It would be better if the authors could include more recent and advanced models like LLAVA-NEXT, etc.






---
[cite 1] Ying, Kaining, et al. MMT-Bench: A Comprehensive Multimodal Benchmark for Evaluating Large Vision-Language Models Towards Multitask AGI. In ICML 2024.

**Questions:**

Do the authors have any plans to expand or adapt MERLIM to include new tasks or modalities as IT-LVLM capabilities evolve?

---

> ### Author Response · Authors · 2024-11-24
>
> We thank reviewer Qhva for the careful reading and overall positive opinion of our submission.
>
> **W1. Overlap with other benchmarks.** As outlined in Lines L139–146, MERLIM overlaps with other benchmarks, such as POPE and MMT-Bench, in tasks like Object Recognition and Counting. However, MERLIM focuses on analyzing hidden hallucinations and tackling more challenging scenarios involving open-ended questions. Unlike traditional benchmarks that assess object recognition through yes/no questions, MERLIM automatically extracts predicted objects from the model's output and compares them to ground truth, enabling a more realistic evaluation. Additionally, MERLIM evaluates the model's performance on both original and edited images, revealing whether its answers are visually grounded. In the inter-object relationship and counting task, we also include the performance for original and edited images and show a decrease in performance in every evaluation. This demonstrates that the hidden hallucination problem extends beyond object recognition to more complex tasks. This highlights a core contribution of our work: hidden hallucinations are a general issue for IT-LVLMs, affecting not just fundamental tasks but also their broader capabilities.
>
> **W2. The field of LVLMs is constantly developing.** Many models evaluated in the paper may not be that strong nowadays. We agree that the field of LVLMs is advancing rapidly. This makes it challenging to evaluate every newly developed IT-LVLM. Nevertheless, we assessed 11 state-of-the-art models to demonstrate the advantages of our benchmark, MERLIM, in highlighting the visual grounding limitations and language biases of IT-LVLMs. We plan to continuously update the leaderboard to include newly released state-of-the-art models, ensuring MERLIM remains a relevant and comprehensive evaluation framework.
>
> **Q2. MERLIM Expansion.** Yes, we see MERLIM as the first step on a larger IT-LVLM Benchmark.  Our core strategy (editing the visual data) allows for direct control of the ground truth, thus allowing us to explore visual arrangements that do not match the regularities of language. We think further visual properties and more complex relationships can be explored, especially with regard to adjectives.

---

### Official Review · Reviewer_1AUQ · 2024-11-04

**Soundness:** 2
**Presentation:** 2
**Contribution:** 2
**Rating:** 5
**Confidence:** 4

**Summary:**

This paper proposes a new dataset  MERLIM to evaluate the zero-shot ability of large vision-language models for fundamental vision tasks, i.e., object recognition, object relationship and object counting. The authors also introduce a inpainting procedure to study the hallucination issues. The authors evaluate the popular LVLMs in the proposed MERLIM.

**Strengths:**

1. The proposed dataset focues on the fundamental object-level  vision tasks for LVLMs, which was seldom explored by previous works.
2. The paper intends to study two types of hallucinations and identify the potential reasons.
3. The authors comprehensively evaluate the popular LVLMs, e.g., BLIP-2, InstructBLIP, LLaVA, in the proposed dataset.

**Weaknesses:**

1. My biggest concern is the data sources all come from the COCO images, which would lead to the bias for LVLMs.
2. The title does not convey the necessary information of the paper. I suggest the authors to add the key motivations in the submission, such as "for evaluating the hallucination in visual objects".
3. The details of inpating process is not well presents. For example, how to choose the edited objects? And how many objects are edited in an image?
4. From Figure 1, it seems that the edited image lose the natural structual in the edited part. Could the authors provide more examples for the edited images?

**Questions:**

1. Some typos should be fixed, e.g. in line 023, "IT-LVMLs" -> "IT-LVLMs"; in line 156, "300.664" -> "300,664"; in line 301, delete "!" and "?". The submission should be carefully checked.
2. What is the version of Qwen-VL. Its performance in Figure 2 and 3 is much lower than BLIP-2, Instruct BLIP, which seems strange.

---

> ### Author Response · Authors · 2024-11-24
>
> We thank 1AUQ for the careful and detailed review of our work, spotting typos and style errors.  They have been corrected on the updated PDF.
>
>
> **W1 MS-COCO data.**  We chose MS-COCO as its overlap with Visual Genome allows us to establish the object relationship task. Additional datasets can be used as input for MERLIM. As long as segmentation data is available, our inpainting method and query strategy can be directly applied, thus generating evaluation data.
> No dataset is completely unbiased thus extending MERLIM to multiple datasets with instance segmentation will still retain some sort of bias from the original dataset. We have already released MERLIM, and we plan on extending it once the performance gap between inpainted and original images begins to close.
>
> **W2 Key motivations in the submission.** Our key motivation is to create a large-scale benchmark that provides a more realistic evaluation scenario for IT-LVLMs. To this end, we create a set of open-ended questions and develop a straightforward way to manipulate the expected outcome of those questions (image inpainting). The result is a flexible and extensible benchmark that shows that IT-LVLMs still have relevant fail-case scenarios in fundamental computer vision tasks. Our flexible analysis also results in the discovery of hidden hallucinations, which are answers that are not effectively visually grounded. A visual grounding error that has not been studied yet but represents a percentage of the performance of IT-LVLMs. We think MERLIM can serve as a tool to advance the area of IT-LVLM by improving the performance in fundamental tasks and better understanding how their performance is affected by text replies that have no effective visual grounding.
>
> **W3 Inpainting Details.** Our inpainting procedure is straightforward, we use COCO masks to define the area to be inpainted. To minimize changes, we remove exactly 1 object from the image. Therefore, we can create multiple inpainted versions of the image (as many as annotated objects are).
> We replace only the pixels in the COCO mask using the pixel data generated by the inpainting method proposed by Li et al. (2022b). No further modifications are made. After inpainting, we apply the quality control described in Appendix A1.
>
> **W4 More inpainted images.** As outlined in the main paper (Line L725), we focus on edited images with a similarity score exceeding 0.7. To further analyze this criterion, we sampled additional images meeting this threshold. Our findings reveal that images with high similarity scores (approximately 0.9) are almost identical to the originals. Even when large objects are removed, the edits remain imperceptible in such cases. Likewise, those images with the lower similarity scores are still highly similar. While some inpainted images may contain minor, localized artifacts (around the edits), these artifacts are subtle. Our quality control process ensures that these artifacts do not alter the appearance of other objects and provide no clues for identifying the removed object.
>
> **Q1.** We have corrected all of them and can be found in the revisited PDF.
>
> **Q2. Qwen-VL performance.** We use Qwen-VL-Chat. Qwen-VL is one of the best-performing IT-VLMs in other benchmarks and is also the best-performing in the Object relationship task (Table 2 curated set). However, the performance in the Object Recognition task is lowered by object hallucinations. Figure 6a) (supplementary material) shows that the precision of Qwen is among the 3 lowest.  Figure 6b) shows that its recall is not too far behind the other models, but questions 1,2, and 5 are particularly bad cases. Overall, we observe Qwen to be very sensitive to the syntax of the question and prone to hallucinations. This combination lowers the F1 score in Figures 2 and 3.

---

> > ### Comment · Reviewer_1AUQ · 2024-11-26
> > **Thanks for the authors' response**
> >
> > Thank the authors for the detailed response. The reply has addressed some of my concerns, including the inpainting details and the performance of Qwen-VL. However, I still think the benchmark has two significant issues, which are also pointed out by other reviewers.
> >
> > (1) It is entirely built upon COCO, which introduces bias and limits the diversity of images for evaluating fundamental vision tasks.
> > (2) The inpainting method lacks the reliability needed to produce real-world quality images.
> >
> > Overall, I feel that the proposed benchmark does not sufficiently stand out from existing datasets in its current form. Therefore, I keep my score as 5.

---

### Official Review · Reviewer_dTrJ · 2024-11-04

**Soundness:** 3
**Presentation:** 3
**Contribution:** 3
**Rating:** 6
**Confidence:** 4

**Summary:**

Authors propose a new benchmark for “IT-LVLMs”, where the main contribution is to remove objects from images and thereby detect “hidden” hallucinations, where the model answer is correct but not grounded in the image. The benchmark tests object recognition, counting, and object relationship. Authors test multiple VLMs, each on several prompts.

**Strengths:**

The paper is well written and understandable, it shows all the relevant details. Missing or incorrect grounding of VLMs is a relevant research question and the approach to simply delete objects and check if the model notices the deletion is clever. Using open-ended answers fits to real use cases of VLMs (compared to testing with multi-choice).

**Weaknesses:**

The work is missing examples. There is only one datapoint shown in figure 1. Please add some randomly picked samples for each of the tasks, to give a better qualitative overview of the dataset.

In figure 1, the tennis racket is still somewhat visible even after deletion, at least the edges of the shape can be detected. However authors verify that an object detector cannot easily find those instances in Appendix A1. Maybe looking at some more edits will show in more details how good this editing works. An alternative approach would be e.g. to use image inpainting but this might lead to other artifacts.

In chapter 3.2 the authors go back to yes/no binary choice questions. This is understandable due to difficulties of posing such questions in an open-ended manner, however asking open questions is more fitting to how a VLM would be used in practice. Maybe it would have been possible to evaluate in a similar manner to 3.1 with making the model list relations and parsing the nouns.

As noted by the authors in line 364 some of the models are already somewhat old, e.g. MiniGPT-4 is ~18 months old at the time of submission. Maybe they could have been replaced by newer models like PaliGemma, or LLaVA 1.6 instead of 1.5. However since the focus is the benchmark this is only a small weakness.

In line 369 consider citing the models with model name instead of author name for clarity.

The citations are messy, e.g. in line 116 they need brackets. In line 201 Li et al. is duplicate. This repeats through the entire work. In general use \cite if the authors are the subject of a sentence, otherwise use \citep to put them into brackets.

In line 443 reference the supplementary material directly by chapter as a clickable reference instead of just as “in the supplementary material”.

The total number of unedited images is somewhat low with less than 10k images, however it’s enough for a benchmark.

In summary I believe if the authors add more clarification and examples, the work is valuable for publication.

**Questions:**

Some details on the output processing could still be added to the supplementary: How exactly are the yes/no answers evaluated, e.g. lowercase, remove dots, then compare to the string “yes”? How are the number answers turned into integers?

Why use only one prompt in 3.3?

Figure 4: Why would models be worse on the edited images? If I understood correctly you mask one of the objects and reduce the count by one. It is not obvious to me why it would be more difficult to count 3 apples after one apple was removed, than to count 4 apples.

---

> ### Author Response · Authors · 2024-11-24
>
> We thank dTrJ for the careful review and favorable view of our work. We are working on the typos and style errors and will notify dTrJ when we update the PDF.
>
> **W1. Examples of tasks.** We have added some example output for each task evaluated by MERLIM in Figure 9 in the revisited PDF supplementary material section (A.6).
>
> **W2 Examples of the edited images.** As outlined in the main paper (Line L723), we focus on edited images with a similarity score exceeding 0.7. To further analyze this criterion, we sampled additional images meeting this threshold (Refer to Figure 5).  Images with high similarity scores (approximately 0.9) are virtually indistinguishable from the originals. Even when large objects are removed, the edits are in line with the image context and are hard to spot. Likewise, the images with the lower similarity score are still highly similar. While some inpainted images may contain localized artifacts, these artifacts are subtle, do not significantly alter the appearance of other objects, and these artifacts provide minimal clues for identifying the removed object.
>
> **W2 Inpainting.** We use the image inpainting method of Li et al. (2022b) (Lines 200-201). Currently, no inpainting method is perfect for completely replacing pixel data with imperceptible and in-context modification. Therefore, we apply the quality control described in Appendix A1.
>
> **W2 Edges of the racquet.** Some minor details of the inpainted object can remain. We identify that some artifacts can also remain at the edge as the polygons from MS-COCO might miss (or over-segment) some pixels at the very edge of the object. Therefore, we perform the YOLO test, described in Appendix A1, to verify the object is no longer recognized in the image.
>
> **W3 open set responses for relation and counting tasks.** We believe our editing and parsing strategy offers significant potential for extension. In future work, we plan to broaden our open-ended question evaluation to account for not only nouns but also adjectives, such as those describing color, position, and other attributes, thereby also evaluating relations between nouns (Relationship task).
>
> **W5,W6,W7. Typos and style.** We have addressed them in the new version of the PDF.
>
> **W8 Number of images.** MERLIM stars with 4.1k original images (MS-COCO val), and from those images we obtain 31K inpainted images (total 35K). Since we have 3 tasks and multiple questions for each individual instruction, we have a total of 300k image-question pairs across 3 tasks. In comparison, hallucination benchmarks like POPE, MMVP, and HallusionBench have between 300 and 500 images (two orders of magnitude less). And contain about 1K to 2K image question pairs (again, two orders of magnitude less). When compared with other hallucination benchmarks, MERLIM is far larger.
>
> **Q1. Details on the output processing.** **For yes/no questions**, we preprocess the answer by converting it to lowercase. If the answer contains the word 'no', it is classified as 'No'. Likewise, if the answer contains 'yes', it is classified as 'Yes.' **For counting questions**, as we outline in L301-302, we prompt the model with the instruction, 'Just answer a number,' allowing us to cast its output using the int() function and evaluate the difference between the prediction and the ground truth. Only answers that can be successfully casted by the cast function' int()' are evaluated. The models produce, on average, 32683.28 valid answers, corresponding to 91.9% of the total questions, considering both edited and original images. We will update the final version to reflect all the details in this process.

---

> ### Author Response · Authors · 2024-11-24
>
> **Q2. Why use only one prompt in 3.3?** As described in Section 3.3 (lines 301-304), we evaluate two prompts. First, we ask the model, 'How many {object category} are there? Just answer the number.' This instructs the model to provide a numerical response, which we cast into an integer using the int() function. Next, we assess the model's congruency with a follow-up prompt, we ask, 'Are there {model answer} {object category}?' or 'Is there one {object category}?' This approach allows us to systematically evaluate the counting ability of the model and if the answer is effectively visually grounded or if it depends on a language bias.
> Clearly, there could be more equivalent language prompts, but MERLIM already captures some language diversity to validate the numerical answers.
>
> **Q3. Why models are, on average, worse on edited images.** This phenomenon is particularly intriguing. The performance gap observed between the original and edited images highlights the spurious visual grounding that is prevalent in most IT-LVLMs. The text output of the IT-LVLMs does not completely correlate with the visual changes in the image, and changes significantly according to the language input. This indicates that IT-LVLMs exhibit limited robustness to modifications in both visual and language inputs. In our paper, MERLIM serves as a diagnostic tool to identify and study this lack of visual grounding and robustness to the input.

---

> ### Comment · Reviewer_dTrJ · 2024-11-25
>
> Thanks for your detailed response. In general I believe the work is meaningful research in the right direction.
>
> However my biggest concern was whether the inpainting actually works well enough. To verify this, I scrolled through the dataset images and found that it can fail quite often. 102331_1 and 107094_0 show cases were humans were only half-erased and still visible. In 100723_3 a person is erased but replaced with meaningless noise. So at least a part of the hallucinations is potentially cases where the object is still visible. These are just two examples but it happens more often, it seems the inpainting AI is not strong enough to reliable produce data of real-world quality.
>
> Therefore, I decided to keep my rating at 6.

---

### Meta-Review · Area_Chair_L2C6 · 2024-12-21

**Metareview:**

The AC appreciates the authors' responses and acknowledges the potential significance of this work. However, the submission falls short in several critical areas. The benchmark relies entirely on the COCO dataset, an prior generation resource with inherent biases that limit its diversity and utility for evaluating fundamental vision tasks (AC agrees). Furthermore, this reliance raises concerns about data leakage, as COCO overlaps with newer datasets and training sets, undermining its relevance to now. The inpainting method is of question, as it is used to create the dataset is also unreliable, often producing artifacts or failing to erase objects, resulting in subpar image quality. Additionally, the evaluation lacks breadth, with a limited range of models tested despite the availability of numerous open-source options. These shortcomings significantly limits the contribution of this work, and the submission does not meet the standards for acceptance.

**Additional Comments On Reviewer Discussion:**

Most reviwers engaged in discussion and provided further justification.

---

### Decision · Program_Chairs · 2025-01-22

Reject